# Unveiling the Sampling Density in Non-Uniform Geometric Graphs

**Raffaele Paolino**
Department of Mathematics &
Munich Center for Machine Learning (MCML),
Ludwig-Maximilians-Universität München
paolino@math.lmu.de

**Aleksandar Bojchevski**
CISPA Helmholtz Center for Information Security
bojchevski@cispa.de

**Stephan Günnemann**
Department of Computer Science &
Munich Data Science Institute,
Technical University of Munich
s.guennemann@tum.de

**Gitta Kutyniok**
Department of Mathematics &
Munich Center for Machine Learning (MCML),
Ludwig-Maximilians-Universität München
kutyniok@math.lmu.de

**Ron Levie**
Faculty of Mathematics,
Technion – Israel Institute of Technology
levieron@technion.ac.il

## Abstract

A powerful framework for studying graphs is to consider them as geometric graphs: nodes are randomly sampled from an underlying metric space, and any pair of nodes is connected if their distance is less than a specified neighborhood radius. Currently, the literature mostly focuses on uniform sampling and constant neighborhood radius. However, real-world graphs are likely to be better represented by a model in which the sampling density and the neighborhood radius can both vary over the latent space. For instance, in a social network communities can be modeled as densely sampled areas, and hubs as nodes with larger neighborhood radius. In this work, we first perform a rigorous mathematical analysis of this (more general) class of models, including derivations of the resulting graph shift operators. The key insight is that graph shift operators should be corrected in order to avoid potential distortions introduced by the non-uniform sampling. Then, we develop methods to estimate the unknown sampling density in a self-supervised fashion. Finally, we present exemplary applications in which the learned density is used to 1) correct the graph shift operator and improve performance on a variety of tasks, 2) improve pooling, and 3) extract knowledge from networks. Our experimental findings support our theory and provide strong evidence for our model.

## 1 Introduction

Graphs are mathematical objects used to represent relationships among entities. Their use is ubiquitous, ranging from social networks to recommender systems, from protein-protein interactions to functional brain networks. Despite their versatility, their non-euclidean nature makes graphs hard to analyze. For instance, the indexing of the nodes is arbitrary, there is no natural definition of orientation, and neighborhoods can vary in size and topology. Moreover, it is not clear how to compare a general pair of graphs since they can have a different number of nodes. Therefore, new ways of thinking about graphs were developed by the community. One approach is proposed in graphon theory (Lovász, 2012): graphs are sampled from continuous graph models called *graphons*, and any two graphs of any size and topology can be compared using certain metrics defined in the space of graphons. A *geometric graph* is an important case of a graph sampled from a graphon. In a geometric graph, a set of points is uniformly sampled from a metric-measure space, and every pair of

points is linked if their distance is less than a specified neighborhood radius. Therefore, a geometric graph inherits a geometric structure from its latent space that can be leveraged to perform rigorous mathematical analysis and to derive computational methods.

Geometric graphs have a long history, dating back to the 60s (Gilbert, 1961). They have been extensively used to model complex spatial networks (Barthelemy, 2011). One of the first models of geometric graphs is the *random geometric graph* (Penrose, 2003), where the latent space is a Euclidean unit square. Various generalizations and modifications of this model have been proposed in the literature, such as *random rectangular graphs* (Estrada & Sheerin, 2015), *random spherical graphs* (Allen-Perkins, 2018), and *random hyperbolic graphs* (Krioukov et al., 2010).

Geometric graphs are particularly useful since they share properties with real-world networks. For instance, random hyperbolic graphs are *small-world*, *scale-free*, with *high clustering* (Papadopoulos et al., 2010; Gugelmann et al., 2012). The small-world property asserts that the distance between any two nodes is small, even if the graph is large. The scale-free property is the description of the degree sequence as a heavy-tailed distribution: a small number of nodes have many connections, while the rest have small neighborhoods. These two properties are related to the presence of *hubs* – nodes with large neighborhoods – while the high clustering is related to the network's community structure.

However, standard geometric graph models focus mainly on uniform sampling, which does not describe real-world networks well. For instance, in location-based social networks, the spatial distribution of nodes is rarely uniform because people congregate around the city centers (Cho et al., 2011; Wang & González, 2009). In online communities such as the LiveJournal social network, non-uniformity arises since the probability of befriending a particular person is inversely proportional to the number of closer people (Hu et al., 2011; Liben-Nowell et al., 2005). In a WWW network, there are more pages for popular topics than obscure ones. In social networks, different demographics (age, gender, ethnicity, etc.) may join a social media platform at different rates. For surface meshes, specific locations may be sampled more finely, depending on the required level of detail.

The imbalance caused by non-uniform sampling could affect the analysis and lead to biased results. For instance, Janssen et al. (2016) show that incorrectly assuming uniform density consistently overestimates the node distances while using the (estimated) density gives more accurate results. Therefore, it is essential to assess the sampling density, which is one of the main goals of this paper.

Barring a few exceptions, non-uniformity is rarely considered in geometric graphs. Iyer & Thacker (2012) study a class of non-uniform random geometric graphs where the radii depend on the location. Martínez-Martínez et al. (2022) study non-uniform graphs on the plane with the density functions specified in polar coordinates. Pratt et al. (2018) consider temporal connectivity in finite networks with non-uniform measures. In all of these works, the focus is on (asymptotic) statistical properties of the graphs, such as the average degree and the number of isolated nodes.

## 1.1 OUR CONTRIBUTION

While traditional Laplacian approximation approaches solve the direct problem – approximating a known continuous Laplacian with a graph Laplacian – in this paper we solve the inverse problem – constructing a graph Laplacian from an observed graph that is guaranteed to approximate an unknown continuous Laplacian. We believe that our approach has high practical significance, as in practical data science on graphs, the graph is typically given, but the underlying continuous model is unknown.

To be able to solve this inverse problem, we introduce the non-uniform geometric graph (NuG) model. Unlike the standard geometric graph model, a NuG is generated by a non-uniform sampling density and a non-constant neighborhood radius. In this setting, we propose a class of graph shift operators (GSOs), called *non-uniform geometric GSOs*, that are computed solely from the topology of the graph and the node/edge features while guaranteeing that these GSOs approximate corresponding latent continuous operators defined on the underlying geometric spaces. Together with Dasoulas et al. (2021) and Sahbi (2021), our work can be listed as a theoretically grounded way to learn the GSO.

Justified by formulas grounded in Monte-Carlo analysis, we show how to compensate for the non-uniformity in the sampling when computing non-uniform geometric GSOs. This requires having estimates both of the sampling density and the neighborhood radii. Estimating these by only observing the graph is a hard task. For example, graph quantities like the node degrees are affected both by the density and the radius, and hence, it is hard to decouple the density from the radius by only observing

the graph. We hence propose methods for estimating the density (and radius) using a self-supervision approach. The idea is to train, against some arbitrary task, a spectral graph neural network, where the GSOs underlying the convolution operators are taken as a non-uniform geometric GSO with learnable density. For the model to perform well, it learns to estimate the underlying sampling density, even though it is not directly supervised to do so. We explain heuristically the feasibility of the self-supervision approach on a sub-class of non-uniform geometric graphs that we call *geometric graphs with hubs*. This is a class of geometric graphs, motivated by properties of real-world networks, where the radius is roughly piece-wise constant, and the sampling density is smooth.

We show experimentally that the NuG model can effectively model real-world graphs by training a graph autoencoder, where the encoder embeds the nodes in an underlying geometric space, and the decoder produces edges according to the NuG model. Moreover, we show that using our non-uniform geometric GSOs with learned sampling density in spectral graph neural networks improves downstream tasks. Finally, we present proof-of-concept applications in which we use the learned density to improve pooling and extract knowledge from graphs.

## 2 NON-UNIFORM GEOMETRIC MODELS

In this section, we define non-uniform geometric GSOs, and a subclass of such GSOs called geometric graphs with hubs. To compute such GSOs from the data, we show how to estimate the sampling density from a given graph using self-supervision.

### 2.1 GRAPH SHIFT OPERATORS AND KERNEL OPERATORS

We denote graphs by $\mathcal{G} = (\mathcal{V}, \mathcal{E})$, where $\mathcal{V}$ is the set of nodes, $|\mathcal{V}|$ is the number of nodes, and $\mathcal{E}$ is the set of edges. A one-dimensional graph signal is a mapping $\mathbf{u} : \mathcal{V} \to \mathbb{R}$. For a higher feature dimension $F \in \mathbb{N}$, a signal is a mapping $\mathbf{u} : \mathcal{V} \to \mathbb{R}^F$. In graph data science, typically, the data comprises only the graph structure $\mathcal{G}$ and node/edge features $\mathbf{u}$, and the practitioner has the freedom to design a graph shift operator (GSO). Loosely speaking, given a graph $\mathcal{G} = (\mathcal{V}, \mathcal{E})$, a GSO is any matrix $\mathbf{L} \in \mathbb{R}^{|\mathcal{V}| \times |\mathcal{V}|}$ that respects the connectivity of the graph, i.e., $L_{i,j} = 0$ whenever $(i, j) \notin \mathcal{E}$, $i \neq j$ (Mateos et al., 2019). GSOs are used in graph signal processing to define filters, as functions of the GSO of the form $f(\mathbf{L})$, where $f : \mathbb{R} \to \mathbb{R}$ is, e.g., a polynomial (Defferrard et al., 2016) or a rational (Levie et al., 2019) function. The filters operate on graph signals $\mathbf{u}$ by $f(\mathbf{L})\mathbf{u}$. Spectral graph convolutional networks are the class of graph neural networks that implement convolutions as filters. When a spectral graph convolutional network is trained, only the filters $f : \mathbb{R} \to \mathbb{R}$ are learned. One significant advantage of the spectral approach is that the convolution network is not tied to a specific graph, but can rather be transferred between different graphs of different sizes and topologies.

In this work, we see GSOs as randomly sampled from kernel operators defined on underlying geometric spaces. The underlying spaces are modelled as metric spaces. To allow modeling the random sampling of points, each metric space is also assumed to be a probability space.

**Definition 1.** Let $(\mathcal{S}, d, \mu)$ be a metric-probability space[1] with probability measure $\mu$ and metric $d$; let $m \in \mathrm{L}^\infty(\mathcal{S})^2$; let $K \in \mathrm{L}^\infty(\mathcal{S} \times \mathcal{S})$. The *metric-probability Laplacian* $\mathcal{L} = \mathcal{L}_{K,m}$ is defined as

$$\mathcal{L} : \mathrm{L}^\infty(\mathcal{S}) \to \mathrm{L}^\infty(\mathcal{S}), \ (\mathcal{L}u)(x) = \int_{\mathcal{S}} K(x,y)\, u(y)\, \mathrm{d}\mu(y) - m(x)\, u(x). \tag{1}$$

For example, let $\mathcal{S}$ be a Riemannian manifold, and take $K(x, y) = \mathbb{1}_{\mathrm{B}_\alpha(x)}(y)/\mu(\mathrm{B}_\alpha(x))$ and $m(x) = 1$, where $\mathrm{B}_\alpha(x)$ is the ball or radius $\alpha$ about $x$. In this case, the operator $\mathcal{L}_{K,m}$ approximates the Laplace-Beltrami operator when $\alpha$ is small (Burago et al., 2019).

A random graph is generated by randomly sampling points from the metric-probability space $(\mathcal{S}, d, \mu)$. As a modeling assumption, we suppose the sampling is performed according to a measure $\nu$. We assume $\nu$ is a weighted measure with respect to $\mu$, i.e., there exists a density function $\rho : \mathcal{S} \to (0, \infty)$

---

[1]A metric-probability space is a triple $(\mathcal{S}, d, \mu)$, where $\mathcal{S}$ is a set of points, and $\mu$ is the Borel measure corresponding to the metric $d$.

[2]A function $g : \mathcal{S} \to \mathbb{R}$ is an element of $\mathrm{L}^\infty(\mathcal{S})$ iff. $\exists M < \infty : \mu(\{x \in \mathcal{S} : |g(x)| > M\}) = 0$. The norm in $\mathrm{L}^\infty(\mathcal{S})$ is the essential supremum, i.e. $\inf\{M \geq 0 : |g(x)| \leq M$ for almost every $x \in \mathcal{S}\}$.

such that $\mathrm{d}\nu(y) = \rho(y)\,\mathrm{d}\mu(y)^3$. We assume that $\rho$ is bounded away from zero and infinity. Using a change of variable, it is easy to see that

$$(\mathcal{L}u)(x) = \int_{\mathcal{S}} K(x,y)\,\rho(y)^{-1}\,u(y)\,\mathrm{d}\nu(y) - m(x)\,u(x)\,.$$

Let $\mathbf{x} = \{x_i\}_{i=1}^N$ be a random independent sample from $\mathcal{S}$ according to the distribution $\nu$. The corresponding sampled GSO $\mathbf{L}$ is defined by

$$L_{i,j} = N^{-1}K(x_i,x_j)\rho(x_j)^{-1} - m(x_i)\,. \tag{2}$$

Given a signal $u \in \mathrm{L}^\infty(\mathcal{S})$, and its sampled version $\mathbf{u} = \{u(x_i)\}_{i=1}^N$, it is well known that $(\mathbf{L}u)_i$ approximates $\mathcal{L}u(x_i)$ for every $i \in \{1,\ldots,N\}$ (Hein et al., 2007; von Luxburg et al., 2008).

## 2.2 Non-Uniform Geometric GSOs

According to (2), a GSO $\mathbf{L}$ can be directly sampled from the metric-probability Laplacian $\mathcal{L}$. However, such an approach would violate our motivating guidelines, since we are interested in GSOs that can be computed directly from the graph structure, without explicitly knowing the underlying continuous kernel and density. In this subsection, we define a class of metric-probability Laplacians that allow such direct sampling. For that, we first define a model of adjacency in the metric space.

**Definition 2.** Let $(\mathcal{S}, d, \mu)$ be a metric-probability space. Let $\alpha : \mathcal{S} \to (0, +\infty)$ be a non-negative measurable function named *neighborhood radius*. The *neighborhood model* $\mathcal{N}$ is defined as the set-valued function that assigns to each $x \in \mathcal{S}$ the ball $\mathcal{N}(x) := \{y \in \mathcal{S} \,:\, d(x,y) \leq \max\big(\alpha(x), \alpha(y)\big)\}$.

Since $y \in \mathcal{N}(x)$ implies $x \in \mathcal{N}(y)$ for all $x, y \in \mathcal{S}$, Def. 2 models only symmetric graphs. Next, we define a class of continuous Laplacians based on neighborhood models.

**Definition 3.** Let $(\mathcal{S}, d, \mu)$ be a metric-probability space, and $\mathcal{N}$ a neighborhood model as in Def. 2. Let $m^{(i)} : \mathbb{R} \to \mathbb{R}$ be a continuous function for every $i \in \{1, \ldots, 4\}$. The *metric-probability Laplacian model* is the kernel operator $\mathcal{L}_\mathcal{N}$ that operates on signals $u : \mathcal{S} \to \mathbb{R}$ by

$$\begin{aligned}
(\mathcal{L}_\mathcal{N}u)(x) := &\int_{\mathcal{N}(x)} m^{(1)}\big(\mu(\mathcal{N}(x))\big)\, m^{(2)}\big(\mu(\mathcal{N}(y))\big)\, u(y)\,\mathrm{d}\mu(y) \\
&- \int_{\mathcal{N}(x)} m^{(3)}\big(\mu(\mathcal{N}(x))\big)\, m^{(4)}\big(\mu(\mathcal{N}(y))\big)\,\mathrm{d}\mu(y)\, u(x)\,.
\end{aligned} \tag{3}$$

In order to give a concrete example, suppose the neighborhood radius $\alpha(x) = \alpha$ is a constant, $m^{(1)}(x) = m^{(3)}(x) = x^{-1}$, and $m^{(2)}(x) = m^{(4)}(x) = 1$, then (3) gives

$$(\mathcal{L}_\mathcal{N}u)(x) = \frac{1}{\mu(\mathrm{B}_\alpha(x))} \int_{\mathrm{B}_\alpha(x)} u(y)\,\mathrm{d}\mu(y) - u(x)\,,$$

which is an approximation of the Laplace-Beltrami operator.

Since the neighborhood model of $\mathcal{S}$ represents adjacency in the metric space, we make the modeling assumption that graphs are sampled from neighborhood models, as follows. First, random independent points $\mathbf{x} = \{x_i\}_{i=1}^N$ are sampled from $\mathcal{S}$ according to the "non-uniform" distribution $\nu$ as before. Then, an edge is created between each pair $x_i$ and $x_j$ if $x_j \in \mathcal{N}(x_i)$, to form the graph $\mathcal{G}$. Now, a GSO can be sampled from a metric-probability Laplacian model $\mathcal{L}_\mathcal{N}$ by (2), if the underlying continuous model is known. However, such knowledge is not required, since the special structure of the metric-probability Laplacian model allows deriving the GSO directly from the sampled graph $\mathcal{G}$ and the sampled density $\{\rho(x_i)\}_{i=1}^N$. Def. 4 below gives such a construction of GSO. In the following, given a vector $\mathbf{u} \in \mathbb{R}^N$ and a function $m : \mathbb{R} \to \mathbb{R}$, we denote by $m(\mathbf{u})$ the vector $\{m(u_i)\}_{i=1}^N$, and by $\mathrm{diag}(\mathbf{u}) \in \mathbb{R}^{N \times N}$ the diagonal matrix with diagonal $\mathbf{u}$.

---

[3]Formally, $\nu$ is absolutely continuous with respect to $\mu$, with Radon-Nykodin derivative $\rho$.

**Definition 4.** Let $\mathcal{G} = (\mathcal{V}, \mathcal{E})$ be a graph with adjacency matrix $\mathbf{A}$; let $\boldsymbol{\rho} : \mathcal{V} \to (0, \infty)$ be a graph signal, referred to as *graph density*. The *non-uniform geometric GSO* is defined to be

$$\mathbf{L}_{\mathcal{G},\boldsymbol{\rho}} \coloneqq N^{-1} \mathbf{D}_{\boldsymbol{\rho}}^{(1)} \mathbf{A}_{\boldsymbol{\rho}} \mathbf{D}_{\boldsymbol{\rho}}^{(2)} - N^{-1} \operatorname{diag} \left( \mathbf{D}_{\boldsymbol{\rho}}^{(3)} \mathbf{A}_{\boldsymbol{\rho}} \mathbf{D}_{\boldsymbol{\rho}}^{(4)} \mathbf{1} \right) , \tag{4}$$

where $\mathbf{A}_{\rho} = \mathbf{A} \operatorname{diag}(\boldsymbol{\rho})^{-1}$ and $\mathbf{D}_{\boldsymbol{\rho}}^{(i)} = \operatorname{diag} \left( m^{(i)} \left( N^{-1} \mathbf{A}_{\boldsymbol{\rho}} \mathbf{1} \right) \right)$.

Def. 4 can retrieve, as particular cases, the usual GSOs, as shown in Tab. 3 in Appendix C. For example, in case of $m^{(1)}(x) = m^{(3)}(x) = x^{-1}$, $m^{(2)}(x) = m^{(4)}(x) = 1$, and uniform sampling $\rho = 1$, (4) leads to the random-walk Laplacian $\mathbf{L}_{\mathcal{G},1} = \mathbf{D}^{-1}\mathbf{A} - \mathbf{I}$. The non-uniform geometric GSO in Def. 4 is the Monte-Carlo approximation of the metric-probability Laplacian in Def. 3. This is shown in the following proposition, whose proof can be found in Appendix D.

**Proposition 1.** *Let $\mathcal{G} = (\mathcal{V}, \mathcal{E})$ be a random graph with i.i.d. sample $\mathbf{x} = \{x_i\}_{i=1}^N$ from the metric-probability space $(\mathcal{S}, d, \mu)$ with neighborhood structure $\mathcal{N}$. Let $\mathbf{L}_{\mathcal{G},\boldsymbol{\rho}}$ be the non-uniform geometric GSO as in Def. 4. Let $u \in \mathrm{L}^\infty(\mathcal{S})$ and $\mathbf{u} = \{u(x_i)\}_{i=1}^N$. Then, for every $i = 1, \ldots, N$,*

$$\mathbb{E} \left( (\mathbf{L}_{\mathcal{G},\boldsymbol{\rho}} \mathbf{u})_i - (\mathcal{L}_{\mathcal{N}} u)(x_i) \right)^2 = \mathcal{O}(N^{-1}). \tag{5}$$

In Appendix D we also show that, in probability at least $1 - p$, it holds

$$\forall\, i \in \{1, \ldots, N\},\ |(\mathbf{L}_{\mathcal{G},\boldsymbol{\rho}} \mathbf{u})_i - (\mathcal{L}_{\mathcal{N}} u)(x_i)| = \mathcal{O} \left( N^{-\frac{1}{2}} \sqrt{\log(1/p) + \log(N)} \right). \tag{6}$$

Prop. 1 means that if we are given a graph that was sampled from a neighborhood model, and we know (or have an estimate of) the sampling density at every node of the graph, then we can compute a GSO according to (4) that is guaranteed to approximate a corresponding unknown metric-probability Laplacian. The next goal is hence to estimate the sampling density from a given graph.

### 2.3 Inferring the Sampling Density

In real-world scenarios, the true value of the sampling density is not known. The following result gives a first rough estimate of the sampling density in a special case.

**Lemma 1.** *Let $(\mathcal{S}, d, \mu)$ be a metric-probability space; let $\mathcal{N}$ be a neighborhood model; let $\nu$ be a weighted measure with respect to $\mu$ with continuous density $\rho$ bounded away from zero and infinity. There exists a function $c : \mathcal{S} \to \mathcal{S}$ such that $c(x) \in \mathcal{N}(x)$ and $(\rho \circ c)(x) = \nu\big(\mathcal{N}(x)\big)/\mu\big(\mathcal{N}(x)\big)$.*

The proof can be found in Appendix D. In light of Lemma 1, if the neighborhood radius of $x$ is small enough, if the volumes $\mu(\mathcal{N}(x))$ are approximately constant, and if $\rho$ does not vary too fast, the sampling density at $x$ is roughly proportional to $\nu(\mathcal{N}(x))$, that is, the likelihood a point is drawn from $\mathcal{N}(x)$. Therefore, in this situation, the sampling density $\rho(x)$ can be approximated by the degree of the node $x$. In practice, we are interested in graphs where the volumes of the neighborhoods $\mu(\mathcal{N}(x))$ are not constant. Still, a normalization of the GSO by the degree can soften the distortion introduced by non-uniform sampling, at least locally in areas where $\mu(\mathcal{N}(x))$ is slowly varying. This suggests that the degree of a node is a good input feature for a method that learns the sampling density from the graph structure and the node features. Such a method is developed next.

### 2.4 Geometric Graphs with Hubs

When designing a method to estimate the sampling density from the graph, the degree is not a sufficient input parameter. The reason is that the degree of a node has two main contributions: the sampling density and the neighborhood radius. The problem of decoupling the two contributions is difficult in the general case. However, if the sampling density is slowly varying, and if the neighborhood radius is piecewise constant, the problem becomes easier. Intuitively, a slowly varying sampling density causes a slight change in the degree of adjacent nodes. In contrast, a sudden change in the degree is caused by a radius jump. In time-frequency analysis and compressed sensing, various results guarantee the ability to separate a signal into its different components, e.g., piecewise constant and smooth components (Do et al., 2022; Donoho & Kutyniok, 2013; Gribonval & Bacry, 2003). This motivates our model of *geometric graphs with hubs*.

**Definition 5.** A *geometric graph with hubs* is a random graph with non-uniform geometric GSO, sampled from a metric-probability space $(\mathcal{S}, d, \mu)$ with neighborhood model $\mathcal{N}$, where the sampling density $\rho$ is Lipschitz continuous in $\mathcal{S}$ and $\mu(\mathcal{N}(x))$ is piecewise constant.

We call this model a geometric graph with hubs since we typically assume that $\mu(\mathcal{N}(x))$ has a low value for most points $x \in \mathcal{S}$, while only a few small regions, called *hubs*, have large neighborhoods. In Section 3.1, we exhibit that geometric graphs with hubs can model real-world graphs. To validate this, we train a graph auto-encoder on real-world networks, where the decoder is restricted to be a geometric graph with hubs. The fact that such a decoder can achieve low error rates suggests that real-world graphs can often be modeled as geometric graphs with hubs.

Geometric graphs with hubs are also reasonable from a modeling point of view. For example, it is reasonable to assume that different demographics join a social media platform at different rates. Since the demographic is directly related to the node features, and the graph roughly exhibits homophily, the features are slowly varying over the graph, and hence, so is the sampling density. On the other hand, hubs in social networks are associated with influencers. The conditions that make a certain user an influencer are not directly related to the features. Indeed, if the node features in a social network are user interests, users that follow an influencer tend to share their features with the influencer, so the features themselves are not enough to determine if a node is deemed to be a center of a hub or not. Hence, the radius does not tend to be continuous over the graph, and, instead, is roughly constant and small over most of the graph (non-influencers), except for some narrow and sharp peaks (influencers).

## 2.5 LEARNING THE SAMPLING DENSITY

In the current section, we propose a strategy to assess the sampling density $\boldsymbol{\rho}$. As suggested by the above discussion, the local changes in the degree of the graph give us a lot of information about the local changes in the sampling density and neighborhood radius of geometric graphs with hubs. Hence, we implement the density estimator as a message-passing graph neural network (MPNN) $\Theta$ because it performs local computations and it is equivariant to node indexing, a property that both the density and the degree satisfy. Since we are mainly interested in estimating the inverse of the sampling density, $\Theta$ takes as input the inverse of the degree and the inverse of the mean degree of the one-hop neighborhood for all nodes in the graph as two input channels.

However, it is not yet clear how to train $\Theta$. Since in real-world scenarios the ground-truth density is not known, we train $\Theta$ in a self-supervised manner. In this context, we choose a task (link prediction, node or graph classification, etc.) on a real-world graph $\mathcal{G}$ and we solve it by means of a graph neural network $\Psi$, referred to as *task network*. Since we want $\Psi$ to depend on the sampling density estimator $\Theta$, we define $\Psi$ as a spectral graph convolution network based on the non-uniform geometric GSO $\mathbf{L}_{\mathcal{G},\Theta(\mathcal{G})}$, e.g., GCN (Kipf & Welling, 2017), ChebNet (Defferrard et al., 2016) or CayleyNet (Levie et al., 2019). We, therefore, train $\Psi$ end-to-end on the given task.

The idea behind the proposed method is that the task depends mostly on the underlying continuous model. For example, in shape classification, the label of each graph depends on the surface from which the graph is sampled, rather than the specific intricate structure of the discretization. Therefore, the task network $\Psi$ can perform well if it learns to ignore the particular fine details of the discretization, and focus on the underlying space. The correction of the GSO via the estimated sampling density ((4)) gives the network exactly such power. Therefore, we conjecture that $\Theta$ will indeed learn how to estimate the sampling density for graphs that exhibit homophily. In order to verify the previous claim, and to validate our model, we focus on link prediction on synthetic datasets (see Appendix B), for which the ground-truth sampling density is known. As shown in Fig. 1, the MPNN $\Theta$ is able to correctly identify hubs, and correctly predict the ground-truth density in a self-supervised manner.

## 3 EXPERIMENTS

In the following, we validate the NuG model experimentally. Moreover, we verify the validity of our method first on synthetic datasets, then on real-world graphs in a transductive (node classification) and inductive (graph classification) setting. Finally, we propose proof-of-concept applications in explainability, learning GSOs, and differentiable pooling.

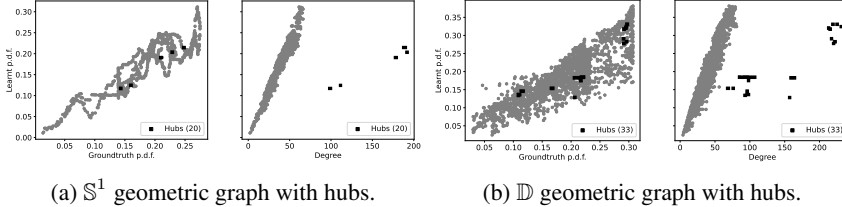

(a) $\mathbb{S}^1$ geometric graph with hubs.  (b) $\mathbb{D}$ geometric graph with hubs.

Figure 1: Example of the learned probability density function in link prediction, where the underlying metric space is (a) the unit-circle, and (b) the unit disk. (Left) Ground-truth sampling density vs. learned sampling density at the nodes. (Right) Degree vs. learned sampling density.

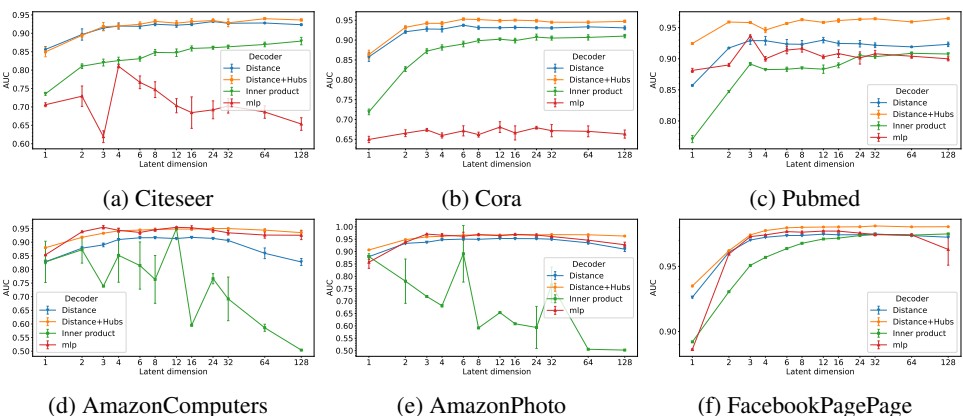

(a) Citeseer  (b) Cora  (c) Pubmed

(d) AmazonComputers  (e) AmazonPhoto  (f) FacebookPagePage

Figure 2: Test AUC for link prediction task as a function of the dimension of the latent space. Performances averaged across 10 runs on each value of the latent dimension.

## 3.1 LINK PREDICTION

The method proposed in Section 2.5 is applied on synthetic datasets of geometric graphs with hubs (see for details Appendices A.1 to A.2). In Fig. 1, it is shown that $\Theta$ is able to correctly predict the value of the sampling density. The left plots of Figs. 1a and 1b show that the density is well approximated both at hubs and non-hubs. Looking at the right plots, it is evident that the density cannot be predicted solely from the degree.

Fig. 2 and Fig. 7 in Appendix A.3 show that the NuG model is able to effectively represent real-world graphs, outperforming other graph auto-encoder methods (see Tab. 1 for the number of parameters of each method). Here, we learn an auto-encoder with four types of decoders: inner product, MLP, constant neighborhood radius, and piecewise constant neighborhood radius corresponding to a geometric graph with hubs (see Appendix A.3 for more details). Better performances are reached if the graph is allowed to be a geometric graph with hubs as in Def. 5. Moreover, the performances of distance and distance+hubs decoder seem to be consistent among different datasets, unlike the inner product and MLP decoders. This corroborates the claims that real-world graphs can be better modeled as geometric graphs with non-constant neighborhood radius. Fig. 8 in Appendix A.3 shows the learned probabilities of being a hub, and the learned values of $\alpha$ and $\beta$, for the Pubmed graph.

## 3.2 NODE CLASSIFICATION

Another exemplary application is to use a non-uniform geometric GSO $\mathbf{L}_{\mathcal{G},\rho}$ (Def. 4) in a spectral graph convolution network for node classification tasks, where the density $\rho_i$ at each node $i$ is computed by a different graph neural network, and the whole model is trained end-to-end on the task. The details are reported in Appendix A.2. In Fig. 3 we show the accuracy of the best-scoring GSO out of the ones reported in Tab. 3 when the density is ignored against the best-scoring GSO when the sampling density is learned. For Citeseer and FacebookPagePage, the best GSOs are the symmetric normalized adjacency matrix. For Cora and Pubmed, the best density-ignored GSO is the symmetric

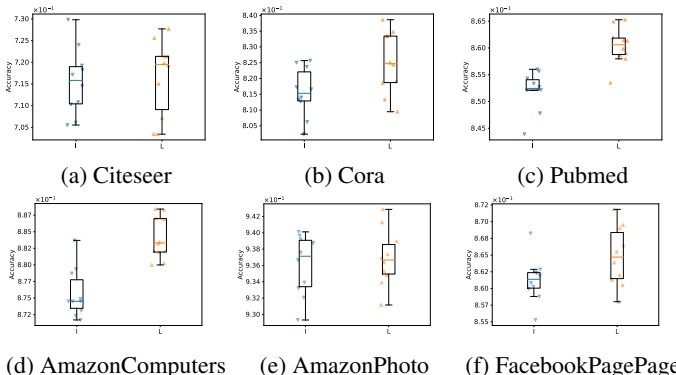

Figure 3: Test accuracy on node classification task. Comparison between the best scoring GSOs when the density is ignored (I) or learned (L). Results averaged across 10 runs: each point represents the performance at one run

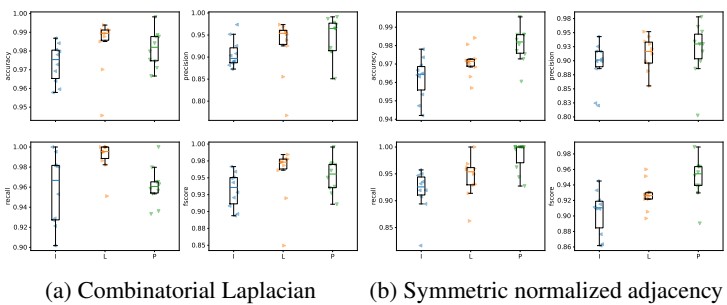

Figure 4: Test metrics of the graph classification task on the AIDS dataset, using the combinatorial Laplacian (a) and the symmetric normalized adjacency (b), averaged over 10 runs. Comparison when the importance $\boldsymbol{\rho}^{-1}$ is ignored (I), used to correct the Laplacian (L), or used for pooling (P). Each point represents the performance at one run. In (a) the best performances are reached when $\boldsymbol{\rho}^{-1}$ is used to correct the Laplacian, and in (b) when $\boldsymbol{\rho}^{-1}$ is used for pooling.

normalized adjacency matrix, while the best density-normalized GSO is the adjacency matrix. For AmazonComputers and AmazonPhoto, the best-scoring GSOs are the symmetric normalized Laplacian. This validates our analysis: if the sampling density is ignored, the best choice is to normalize the Laplacian by the degree to soften the distortion of non-uniform sampling.

## 3.3 GRAPH CLASSIFICATION & DIFFERENTIABLE POOLING

In this experiment we perform graph classification on the AIDS dataset (Riesen & Bunke, 2008), as explained in Appendix A.2. Fig. 4 shows that the classification performances of a spectral graph neural network are better if a quota of parameters is used to learn $\boldsymbol{\rho}$ which is used in a non-uniform geometric GSO (Def. 4). The learnable $\boldsymbol{\rho}$ on the AIDS dataset can be used not only to correct the Laplacian but also to perform a better pooling (see Appendix A.2 for the details). Usually, a graph convolutional neural network is followed by a global pooling layer in order to extract a representation of the whole graph. A vanilla pooling layer aggregates uniformly the contribution of all nodes. We implemented a weighted pooling layer that takes into account the importance of each node. As shown in Fig. 4, the weighted pooling layer can indeed improve performances on the graph classification task. Fig. 6 in Appendix A.2 shows a comparison between the degree, the density learned to correct the GSO and the density learned for pooling. From the plot it is clear that the degree cannot predict the density. Indeed, the sampling density at nodes with the same degree can have different values .

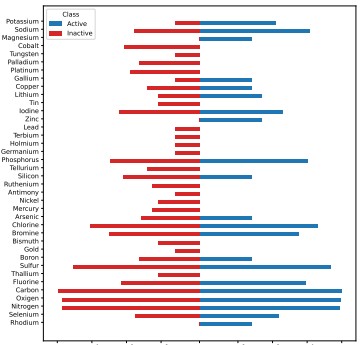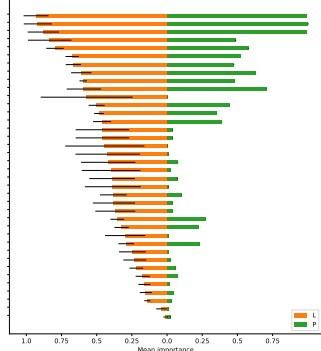

Figure 5: (Left) Distribution of chemical elements per class (active, inactive respectively in blue, red) computed as the number of compounds labeled as active (inactive) containing that particular element divided by the number of active (inactive) compounds. This is a measure of rarity. For example, potassium is present in $5$ out of $400$ active compounds, and in $1$ over $1600$ inactive compounds. Hence, it is more rare to find potassium in an inactive compound. (Right) The mean importance of each element when $\boldsymbol{\rho}^{-1}$ is used to correct the GSO (L, orange) and when it is used for weighted pooling (P, green). Carbon, oxygen, and nitrogen have low mean importance, which makes sense as they are present in almost every compound, as shown in the left plot. The chemical elements are sorted according to their mean importance when $\boldsymbol{\rho}^{-1}$ is used to correct the GSO (orange bars).

### 3.4 EXPLAINABILITY IN GRAPH CLASSIFICATION

In this experiment, we show how to use the density estimator for explainability. The inverse density vector $\boldsymbol{\rho}^{-1}$ can be interpreted as a measure of importance of each node, relative to the task at hand, instead of sampling density. Thinking about $\boldsymbol{\rho}^{-1}$ as importance is useful when the graph is not naturally seen as randomly generated from a graphon model. We applied this paradigm to the AIDS dataset, as explained in the previous subsection. The better classification performances when $\boldsymbol{\rho}$ is learned demonstrates that $\boldsymbol{\rho}$ is an important feature for the classification task, and hence, it can be exploited to extract knowledge from the graph. We define the mean importance of each chemical element $e$ as the sum of all values of $\boldsymbol{\rho}^{-1}$ corresponding to nodes labeled as $e$ divided by the number of nodes labeled $e$. Fig. 5 shows the mean importance of each element, when $\boldsymbol{\rho}^{-1}$ is estimated by using it as a module in the task network in two ways. (1) The importance $\boldsymbol{\rho}^{-1}$ is used to correct the GSO. (2) The importance $\boldsymbol{\rho}^{-1}$ is used in a pooling layer, that maps the output of the graph neural network $\Psi$ to one feature of the form $\sum_{j=1}^{|\mathcal{V}|} \rho_j^{-1} \Psi(\mathbf{X})_j$, where $\mathbf{X}$ denotes the node features. In both cases, the most important elements are the same; therefore, the two methods seem to be consistent.

### CONCLUSIONS

In this paper, we addressed the problem of learning the latent sampling density by which graphs are sampled from their underlying continuous models. We developed formulas for representing graphs given their connectivity structure and sampling density using non-uniform geometric GSOs. We then showcased how the density of geometric graphs with hubs can be estimated using self-supervision, and validated our approach experimentally. Last, we showed how knowing the sampling density can help with various tasks, e.g., improving spectral methods, improving pooling, and gaining knowledge from graphs. One limitation of our methodology is the difficulty in validating that real-world graphs are indeed sampled from latent geometric spaces. While we reported experiments that support this modeling assumption, an important future direction is to develop further experiments and tools to support our model. For instance, can we learn a density estimator on one class of graphs and transfer it to another? Can we use ground-truth demographic data to validate the estimated density in social networks? We believe future research will shed light on those questions and find new ways to exploit the sampling density for various applications.

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

# A IMPLEMENTATION DETAILS

## A.1 SYNTHETIC DATASET GENERATION

This section explains how to generate a synthetic dataset of geometric graphs with hubs. We first consider a metric space. For our experiments, we mainly focused on the unit-circle $\mathbb{S}^1$ and on the unit-disk $\mathbb{D}$ (see Appendix B for more details). Each graph is generated as follows. First, a non-uniform distribution is randomly generated. We considered an angular non-uniformity as described in Def. 6, where the number of oscillating terms, as well as the parameters $\boldsymbol{c}, \boldsymbol{n}, \boldsymbol{\mu}$, are chosen randomly. In the case of 2-dimensional spaces, the radial distribution is the one shown in Tab. 2. According to each generated probability density function, $N$ points $\{x_i\}_{i=1}^N$ are drawn independently. Among them, $m < N$ are chosen randomly to be hubs, and any other node whose distance from a hub is less than some $\varepsilon > 0$ is also marked as a hub. We consider two parameters $\alpha, \beta > 0$. The neighborhood radius about non-hub (respectively, hubs) nodes is taken to be $\alpha$ (respectively $\alpha + \beta$). Any two points are then connected if

$$d(x_i, x_j) \leq \max\{r(x_i), r(x_j)\}, \; r(x) = \begin{cases} \alpha & x \text{ is non-hub} \\ \alpha + \beta & x \text{ is hub} \end{cases}.$$

In practical terms, $\alpha$ is computed such that the resulting graph is strongly connected, hence, it differs from graph to graph; $\beta$ is set to be $3\alpha$ and $\epsilon$ to be $\alpha/10$.

## A.2 DENSITY ESTIMATION WITH SELF-SUPERVISION

**Density Estimation Network** In our experiments, the inverse of the sampling density, $1/\rho$, is learned by means of an EdgeConv neural network $\Theta$ (Wang et al., 2019), which is referred to as PNet in the following, where the message function is a multi-layer perceptron (MLP), and the aggregation function is $\max(\cdot)$, followed by a $\mathrm{abs}(\cdot)$ non-linearity. The number of hidden layers, hidden channels, and output channels is 3, 32, and 1, respectively. Since the degree is an approximation of the sampling density, as stated in Lemma 1, and since we are interested in computing its inverse to correct the GSO, the input of PNet is the inverse of the degree and the inverse of the mean degree of the one-hop neighborhood. Justified by the Monte-Carlo approximation

$$1 = \int_{\mathcal{S}} \mathrm{d}\mu(y) = \int_{\mathcal{S}} \rho(y)^{-1} \, \mathrm{d}\nu(y) \approx N^{-1} \sum_{i=1}^N \rho(x_i)^{-1}, \; x_i \sim \rho \; \forall i = 1, \ldots, N,$$

the output of PNet is normalized by its mean.

**Self-Supervision of PNet via Link Prediction on Synthetic Dataset** To train the PNet $\Theta$, for each graph $\mathcal{G}$, we use $\Theta(\mathcal{G})$ to define a GSO $\mathbf{L}_{\mathcal{G}, \Theta(\mathcal{G})}$. Then, we define a graph auto-encoder, where the encoder is implemented as a spectral graph convolution network with GSO $\mathbf{L}_{\mathcal{G}, \Theta(\mathcal{G})}$. The decoder is the usual inner-product decoder. The graph signal is a slice of 20 random columns of the adjacency matrix. The number of hidden channels, hidden layers, and output channels is respectively 32, 2, and 2. For each node $j$, the network outputs a feature $\Theta(\mathcal{G})_j$ in $\mathbb{R}^n$. Here, $\mathbb{R}^n$ is seen as the metric space underlying the NuG. In our experiments (Section 3.1), we choose $n = 2$. Some results are shown in Fig. 1.

**Node Classification** Let $\mathcal{G}$ be the real-world graph. In Section 3.2, we considered $\mathcal{G}$ to be one of the graphs reported in Tab. 1. The task network $\Psi$ is a polynomial convolutional neural network implementing a GSO $\mathbf{L}_{\mathcal{G}, \Theta(\mathcal{G})}$, where $\Theta$ is the PNet; the order of the polynomial spectral filters is 1, the number of hidden channels 32, and the number of hidden layers 2; the GSOs used are the ones in Tab. 3. The optimizer is ADAM (Kingma & Ba, 2015) with learning rate $10^{-2}$. We split the nodes in training (85%), validation (5%), and test (10%) in a stratified fashion, and apply early stopping. The performances of the method are shown in Fig. 3.

**Graph Classification** Let $\mathcal{G}$ be the real-world graph. In Section 3.4, $\mathcal{G}$ is any compound in the AIDS dataset. The task network $\Psi$ is a polynomial convolutional neural network implementing a GSO $\mathbf{L}_{\mathcal{G}, \Theta(\mathcal{G})}$, where $\Theta$ is the PNet; the order of the spectral polynomial filters is 1, the number of

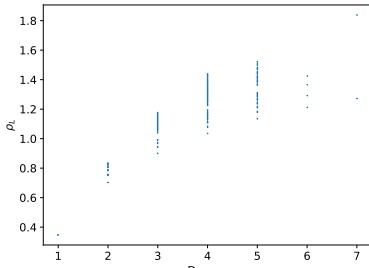 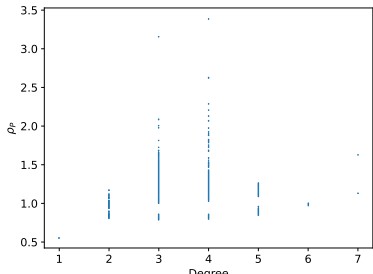

Figure 6: Comparison between degree, density learnt to correct the GSO $\rho_L$, and density learnt to perform weighted pooling $\rho_P$, AIDS dataset.

hidden channels 128 and the number of hidden layers 2. The optimizer is ADAM with learning rate $10^{-2}$. We perform a stratified splitting of the graphs in training (85%), validation (5%), and test (10%), and applied early stopping. The chosen batch size is 64. The pooling layer is a global add layer.

In case of weighted pooling as in Section 3.3, the task network $\Psi$ implements as GSO $\mathbf{L}_{\mathcal{G},\mathbf{1}}$, while $\Theta$ is used to output the weights of the pooling layer. The performance metrics of both approaches are shown in Fig. 4

### A.3 GEOMETRIC GRAPHS WITH HUBS AUTO-ENCODER

Here, we validate that real-world graphs can be modeled approximately as geometric graphs with hubs, as claimed in Section 3.1. We consider the datasets listed in Tab. 1. The auto-encoder is defined as follows. Let $\mathcal{G}$ be the real-world graph with $N$ nodes and $F$ node features; let $\mathbf{X} \in \mathbb{R}^{N \times F}$ be the feature matrix. Let $n$ be the dimension of the metric space in which nodes are embedded. Let $\Psi$ be a spectral graph convolutional network, referred to as *encoder*. Let $\Psi(\mathbf{X})_i$ and $\Psi(\mathbf{X})_j \in \mathbb{R}^n$ be the embedding of nodes $i$ and $j$ respectively. A decoder is a mapping $\mathbb{R}^n \times \mathbb{R}^n \to [0,1]$ that takes as input the embedding of two nodes $i$, $j$ and returns the probability that the edge $(i,j)$ exists.

We use four types of decoders. (1) The *inner product decoder* from Kipf & Welling (2016) is defined as $\sigma\left(\langle\Psi(\mathbf{X})_i, \Psi(\mathbf{X})_j\rangle\right)$, where $\sigma(\cdot)$ is the logistic sigmoid function. (2) The *MLP decoder* is defined as $\sigma\left(\mathrm{MLP}([\Psi(\mathbf{X})_i, \Psi(\mathbf{X})_j])\right)$, where $[\Psi(\mathbf{X})_i, \Psi(\mathbf{X})_j] \in \mathbb{R}^{2n}$ denotes the concatenation of $\Psi(\mathbf{X})_i$ and $\Psi(\mathbf{X})_j$, and MLP denotes a multi-layer perceptron. (3) The *distance decoder* corresponds to geometric graphs. It is defined as $\sigma(\alpha - \|\Psi(\mathbf{X})_i - \Psi(\mathbf{X})_j\|_2)$, where $\alpha$ is the trainable neighborhood radius. (4) The *distance+hubs decoder* corresponds to geometric graphs with hubs. It is defined as $\sigma(\alpha + \max\{\Upsilon(\tilde{\mathbf{D}})_i, \Upsilon(\tilde{\mathbf{D}})_j\}\beta - \|\Psi(\mathbf{X})_i - \Psi(\mathbf{X})_j\|_2)$, where $\alpha$, $\beta$ are trainable parameters that describe the radii of hubs and non-hubs. $\Upsilon$ is a message-passing graph neural network (with the same architecture of PNet) that takes as input a signal $\tilde{\mathbf{D}}$ computed from the node degrees (i.e., the inverse of the degree and the inverse of the mean degree of the one-hop neighborhood), and outputs the probability that each node is a hub. $\Upsilon$ is learned end-to-end together with the rest of the auto-encoder. In order to guarantee that $0 \leq \Upsilon(\mathcal{G})_j \leq 1$ the network is followed by a min-max normalization.

The distance decoder is justified by the fact that the condition $\|\Psi(\mathbf{X})_i - \Psi(\mathbf{X})_j\|_2 \leq \alpha$ can be rewritten as $\mathrm{H}(\alpha - \|\Psi(\mathbf{X})_i - \Psi(\mathbf{X})_j\|_2)$, where $\mathrm{H}(\cdot)$ is the Heaviside function. The Heaviside function is relaxed to the logistic sigmoid for differentiability. Similar reasoning lies behind the formula of the distance+hubs decoder.

The encoder $\Psi$ is a polynomial spectral graph convolutional neural network implementing as GSO the symmetric normalized adjacency matrix; the order of the polynomial filters is 1, the number of hidden channels 32 and the number of hidden layers 2. In the case of inner-product, MLP and distance decoder, the loss is the cross entropy of existing and non-existing edges. In the case of distance+hubs-decoder, we also add $\|\Upsilon(\mathcal{G})\|_1/N$ to the loss, as a regularization term, since in our model we suppose the number of hubs is low. The optimizer is ADAM with learning rate $10^{-2}$. We split the edges in training (85%), validation (5%) and test (10%), and apply early stopping. The

Table 1: Real-world networks used for the link prediction task: graph statistics and number of parameters of the auto-encoder for each of the three decoder types: inner product, distance, distance+hubs and MLP. Since the number of input channels of the MLP decoder depends on the latent dimension $n$, we report the number of parameter for $n = 3$.

| Dataset | Statistics | | | | Decoder | | | |
|---|---|---|---|---|---|---|---|---|
| | N. nodes | N. edges | N. features | N. classes | Inner product | Distance | Distance+Hubs | MLP |
| Citeseer (Yang et al., 2016) | 3,327 | 9,104 | 3,703 | 6 | 237,154 | 237,155 | 239,689 | 239,750 |
| Cora (Yang et al., 2016) | 2,708 | 10,556 | 1,433 | 7 | 91,874 | 91,875 | 94,409 | 94,470 |
| Pubmed (Yang et al., 2016) | 19,717 | 88,648 | 500 | 3 | 32,162 | 32,163 | 34,697 | 34,758 |
| Amazon Computers (Shchur et al., 2019) | 13,752 | 491,722 | 767 | 10 | 49,250 | 49,251 | 51,785 | 51,846 |
| Amazon Photo (Shchur et al., 2019) | 7,650 | 238,162 | 745 | 8 | 47,842 | 47,843 | 50,377 | 50,438 |
| FacebookPagePage (Rozemberczki et al., 2021) | 22,470 | 342,004 | 128 | 4 | 8,354 | 8,355 | 10,889 | 10,950 |

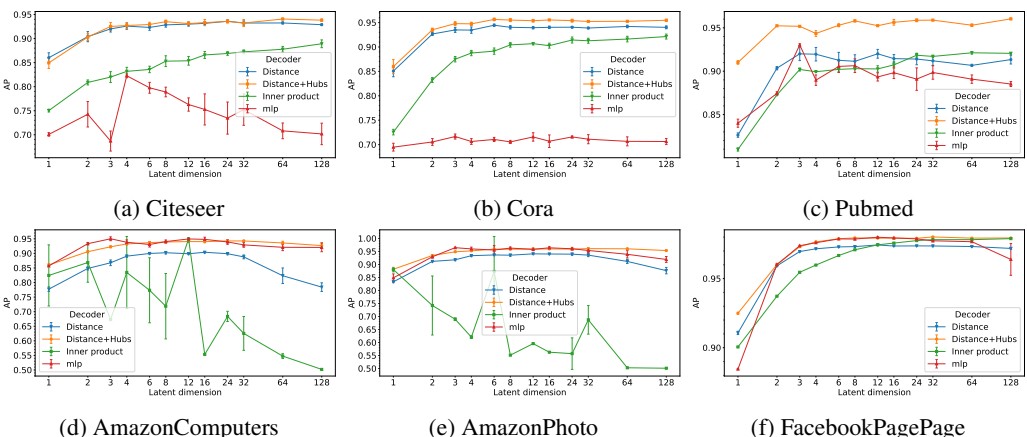

Figure 7: Test AP for link prediction task as a function of the dimension of the latent space. Performances averaged across 10 runs on each value of the latent dimension.

performances of both methods are shown in Figs. 2 and 7, while examples of embeddings and learned $\alpha$ and $\beta$ are shown in Fig. 8.

The distance decoder has one learnable parameter more than the inner-product decoder. Since PNet has a fixed number of input channels, the distance+hubs decoder has $2,535$ learnable parameters more than the inner-product one. On the contrary, the mlp decoder has a number of input channels that depends on the latent dimension; therefore, the number of hidden channels is chosen to guarantee that the number of learnable parameters of the mlp decoder is approximately $2,535$.

## B  SYNTHETIC DATASETS - A BLUEPRINT

In the following, we consider some simple latent metric spaces and construct methods for randomly generating non-uniform samples. For each space, structural properties of the corresponding NuG are studied, such as the expected degree of a node and the expected average degree, in case the radius is fixed and the sampling is non-uniform. All proofs can be found in Appendix D, if not otherwise stated.

Three natural metric measure spaces are the euclidean, spherical, and hyperbolic spaces. If we restrict the attention to 2-dimensional spaces, a way to uniformly sample is summarized in Tab. 2. In all three cases, the *radial* component arises naturally from the measure of the space. A possible way to introduce non-uniformity is changing the *angular* distribution. In this way, preferential directions will be identified, leading to an anisotropic model.

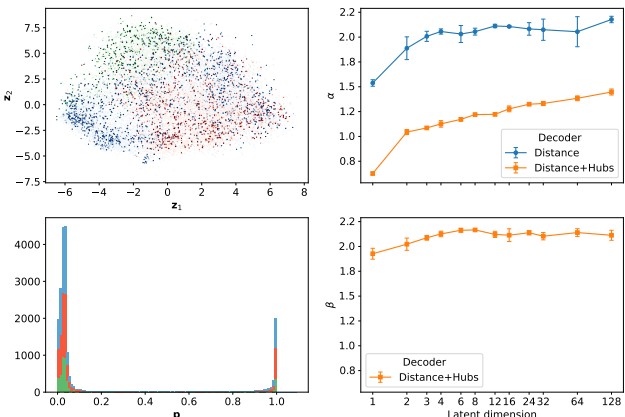

Figure 8: (Top left) Embedding of Pubmed in 2 dimensions using a distance+hubs decoder. The intensity of the color for each node $i$ is proportional to the probability $p_i = \Upsilon(\mathcal{G})_i$ of being a hub. The three colours (red, green and blue) corresponds to the three different classes to which a node can belong, as reported in Tab. 1. (Bottom left) Histogram of the probabilities $\mathbf{p} = \Upsilon(\mathcal{G})$ of being hub divided per class. (Right) Learned values of the radius parameters $\alpha$ (top) and $\beta$ (bottom) of the geometric graph with hubs auto-encoder on Pubmed, as a function of the latent dimension. Results averaged across 10 runs for each value of the latent dimension. The average probability of being a hub is 19.06%, and the number of nodes with a probability of being a hub greater than 0.99 is 10.10%.

Table 2: Properties of euclidean, spherical and hyperbolic spaces of dimension 2. In the case of euclidean and hyperbolic spaces, the uniform distribution refers to a disk of radius $R$.

| Property | Geometry | |
|---|---|---|
| Measure of a ball of radius $\alpha$ | euclidean | $\pi \, \alpha^2$ |
| | spherical | $2 \, \pi \, (1 - \cos(\alpha))$ |
| | hyperbolic | $2 \, \pi \, (\cosh(\alpha) - 1)$ |
| Uniform p.d.f. | euclidean | $(2 \, \pi)^{-1} \mathbb{1}_{[-\pi,\pi)}(\theta) 2 R^{-2} r \, \mathbb{1}_{[0,R)}(r)$ |
| | spherical | $(2 \, \pi)^{-1} \mathbb{1}_{[-\pi,\pi)}(\theta) 2^{-1} \sin(\varphi) \mathbb{1}_{[0,\pi)}(\varphi)$ |
| | hyperbolic | $(2 \, \pi)^{-1} \mathbb{1}_{[-\pi,\pi)}(\theta)(\cosh(R) - 1)^{-1} \sinh(r) \mathbb{1}_{[0,R)}(r)$ |
| Distance in polar coordinates | euclidean | $\sqrt{r_1^2 + r_2^2 - 2 r_1 r_2 \cos(\theta_1 - \theta_2)}$ |
| | spherical | $\arccos\left(\cos(\phi_1)\,\cos(\phi_2) + \sin(\phi_1)\,\sin(\phi_2)\,\cos(\theta_1 - \theta_2)\right)$ |
| | hyperbolic | $\operatorname{arccosh}(\cosh(r_1)\cosh(r_2) - \sinh(r_1)\sinh(r_2)\cos(\theta_1 - \theta_2))$ |

**Definition 6.** Given a natural number $C \in \mathbb{N}$, and vectors $\boldsymbol{c} \in \mathbb{R}^C, \boldsymbol{n} \in \mathbb{N}^C, \boldsymbol{\mu} \in \mathbb{R}^C$ the function

$$\mathrm{sb}(\theta; \boldsymbol{c}, \boldsymbol{n}, \boldsymbol{\mu}) = \frac{1}{B} \sum_{i=1}^{C} c_i \cos(n_i (\theta - \mu_i)) + \frac{A}{B}, \ A = \sum_{i=1}^{C} |c_i|, \ B = 2\pi \left( \sum_{i=1}^{C} |c_i| + \sum_{i:n_i=0} c_i \right),$$

is a continuous, $2\pi$-periodic probability density function. It will be referred to as *spectrally bounded*.

The cosine can be replaced by a generic $2\pi$-periodic function; the only change in the construction will be the offset and the normalization constant.

**Definition 7.** Given a natural number $C \in \mathbb{N}$, and the vectors $\boldsymbol{c} \in \mathbb{R}^C, \boldsymbol{n} \in \mathbb{N}^C, \boldsymbol{\mu} \in \mathbb{R}^C, \boldsymbol{\kappa} \in \mathbb{R}^C_{\geq 0}$, the function

$$\mathrm{mvM}(\theta; \boldsymbol{c}, \boldsymbol{n}, \boldsymbol{\mu}, \boldsymbol{\kappa}) = \frac{1}{B} \sum_{i=1}^{C} c_i \frac{\exp(\kappa_i \cos(n_i (\theta - \mu_i)))}{2\pi \, \mathrm{I}_0(\kappa_i)} + \frac{A}{B},$$

where

$$A = \sum_{i:c_i<0} c_i \frac{\exp(\kappa_i)}{2\pi \, \mathrm{I}_0(\kappa_i)}, \ B = \sum_{i:n_i\geq 1} c_i + \sum_{i:n_i=0} c_i \frac{\exp(\kappa_i)}{\mathrm{I}_0(\kappa_i)} + \sum_{i:c_i<0} c_i \frac{\exp(\kappa_i)}{\mathrm{I}_0(\kappa_i)},$$

is a continuous, $2\pi$-periodic probability density function. It will be referred to as *multimodal von Mises*.

Both densities introduced previously can be thought of as functions over the unit circle. Hence, the very first space to be studied is $\mathbb{S}^1 = \{\mathbf{x} \in \mathbb{R}^2 : \|x\| = 1\}$ equipped with geodesic distance. As shown in the next proposition, the geodesic distance can be computed in a fairly easy way.

**Proposition 2.** *Given two points $\boldsymbol{x}, \boldsymbol{y} \in \mathbb{S}^1$ corresponding to the angles $x, y \in [-\pi, \pi)$, their geodesic distance is equal to*

$$d(\boldsymbol{x}, \boldsymbol{y}) = \pi - |\pi - |x - y||.$$

The next proposition computes the degree of a node in a non-uniform unit circle graph.

**Proposition 3.** *Given a spectrally bounded probability density function as in Def. 6, the expected degree of a node $\theta$ in a unit circle geometric graph with neighborhood radius $\alpha$ is*

$$\deg(\theta) = \frac{2N}{B} \left( \sum_{i:n_i\neq 0} \frac{c_i}{n_i} \cos(n_i (\theta - \mu_i)) \sin(n_i \alpha) + \left( \sum_{i:n_i=0} c_i + A \right) \alpha \right),$$

*and the expected average degree of the whole graph is*

$$\mathbb{E}[\deg(\theta)] = \frac{2\pi N \alpha}{B^2} \left( \sum_{i:n_i\neq 0} \sum_{j:n_i=n_j} c_i c_j \, \cos(n_i (\mu_i - \mu_j)) \frac{\sin(n_i \alpha)}{n_i \alpha} + 2 \left( \sum_{i:n_i=0} c_i + A \right)^2 \right).$$

As a direct consequence, in the limit of $r$ going to zero

$$\lim_{\alpha \to 0^+} \frac{\mathbb{P}[\mathrm{B}_\alpha(\theta)]}{2\alpha} = \frac{1}{B} \left( \sum_{i:n_i\neq 0} c_i \cos(n_i (\theta - \mu_i)) \left( \lim_{\alpha \to 0^+} \frac{\sin(n_i \alpha)}{n_i \alpha} \right) + \left( \sum_{i:n_i=0} c_i + A \right) \right)$$

$$= \mathrm{sb}(\theta; \boldsymbol{c}, \boldsymbol{n}, \boldsymbol{\mu})$$

thus, for sufficiently small $\alpha$, the probability of a ball centered at $\theta$ is proportional to the density computed in $\theta$. Moreover, the error can be computed as

$$\left| \mathrm{sb}(\theta; \boldsymbol{c}, \boldsymbol{n}, \boldsymbol{\mu}) - \frac{\mathbb{P}[\mathrm{B}_\alpha(\theta)]}{2\alpha} \right| \leq \frac{1}{6B} \left( \sum_{i=1}^{C} n_i^2 c_i \right) \alpha^2,$$

which shows that the approximation worsens the more oscillatory terms there are. In the case of multimodal von Mises distribution, a closed formula for the probability of balls does not exist. The following proposition introduces an approximation based solely on cosine functions.

**Proposition 4.** *A multimodal von Mises probability density function can be approximated by a spectrally bounded one.*

The previous result, combined with Prop. 3, gives a way to approximate the expected degree of spatial networks sampled accordingly to a multimodal von Mises angular distribution. However, the computation is straightforward when $\boldsymbol{n}$ is the constant vector $n\,\mathbf{1}$, since the product of two von Mises pdf is the kernel of a von Mises pdf

$$\frac{\exp(\boldsymbol{\kappa}_1\cos(n(\theta-\boldsymbol{\mu}_1)))}{2\,\pi\,\mathrm{I}_0(\boldsymbol{\kappa}_1)}\frac{\exp(\boldsymbol{\kappa}_2\cos(n(\theta-\boldsymbol{\mu}_2)))}{2\,\pi\,\mathrm{I}_0(\boldsymbol{\kappa}_2)}$$

$$=\frac{\exp\left(\sqrt{\boldsymbol{\kappa}_1^2+\boldsymbol{\kappa}_2^2+2\boldsymbol{\kappa}_1\boldsymbol{\kappa}_2\cos(n(\boldsymbol{\mu}_1-\boldsymbol{\mu}_2))}\cos\left(n(\theta-\varphi)\right)\right)}{4\,\pi^2\,\mathrm{I}_0(\boldsymbol{\kappa}_1)\,\mathrm{I}_0(\boldsymbol{\kappa}_2)},$$

where

$$\varphi=n^{-1}\,\arctan\left(\frac{\boldsymbol{\kappa}_1\sin(n\,\boldsymbol{\mu}_1)+\boldsymbol{\kappa}_2\sin(n\,\boldsymbol{\mu}_2)}{\boldsymbol{\kappa}_1\cos(n\,\boldsymbol{\mu}_1)+\boldsymbol{\kappa}_2\cos(n\,\boldsymbol{\mu}_2)}\right).$$

The unit circle model is preparatory to the study of more complex spaces, for instance, the unit disk $\mathbb{D}=\{\mathbf{x}\in\mathbb{R}^2:\|\mathbf{x}\|\leq 1\}$ equipped with geodesic distance, as in Tab. 2.

**Proposition 5.** *Given a spectrally bounded angular distribution as in Def. 6, the degree of a node $(r,\theta)$ in a unit disk geometric graph with neighborhood radius $\alpha$ is*

$$\deg(r,\theta)\approx 2\,\pi\,\alpha^2\,N\,\mathrm{sb}(\theta;\boldsymbol{c},\boldsymbol{n},\boldsymbol{\mu}),$$

*and the average degree of the whole network is*

$$\mathbb{E}[\deg(r,\theta)]\approx\frac{2\,\pi^2\,\alpha^2\,N}{B^2}\left(\sum_{i:n_i\neq 0}\sum_{j:n_i=n_j}c_ic_j\,\cos(n_i\,(\mu_i-\mu_j))+2\left(\sum_{i:n_i=0}c_i+A\right)^2\right).$$

Fig. 9a shows some examples of non-uniform sampling of the unit disk. The last example will be the hyperbolic disk with radius $R\gg 1$, equipped with geodesic distance as in Tab. 2.

**Proposition 6.** *Given a spectrally bounded angular distribution as in Def. 6, the degree of a node $(r,\theta)$ in a hyperbolic geometric graph with neighborhood radius $\alpha$ is*

$$\deg(r,\theta)\approx 8\,N\,e^{\frac{\alpha-R-r}{2}}\,\mathrm{sb}(\theta;\boldsymbol{c},\boldsymbol{n},\boldsymbol{\mu}),$$

*and the average degree of the whole network is $\mathcal{O}(N\,e^{\frac{\alpha-2R}{2}})$.*

The proof can be found in Appendix D. The computed approximation is in line with the findings of Krioukov et al. (2010), where a closed formula for the uniform case is provided when $\alpha=R$. To the best of our knowledge, this is the first work that considers $\alpha\neq R$. Examples of non-uniform sampling of the hyperbolic disk are shown in Fig. 9b.

## C  RETRIEVING AND BUILDING GSOs

In the current section, we first show how to retrieve the usual definition of graph shift operators from Def. 4, and then how Def. 4 can be used to create novel GSOs. For simplicity, for both goals we suppose uniform sampling $\boldsymbol{\rho}=\mathbf{1}$; (4) can be rewritten as

$$\mathbf{L}_{\mathcal{G},\mathbf{1}}=N^{-1}\,\mathrm{diag}\left(m^{(1)}\left(N^{-1}\mathbf{d}\right)\right)\mathbf{A}\,\mathrm{diag}\left(m^{(2)}\left(N^{-1}\mathbf{d}\right)\right)$$

$$-N^{-1}\,\mathrm{diag}\left(\mathrm{diag}\left(m^{(3)}\left(N^{-1}\mathbf{d}\right)\right)\mathbf{A}\,\mathrm{diag}\left(m^{(4)}\left(N^{-1}\mathbf{d}\right)\right)\mathbf{1}\right) \tag{7}$$

where $\mathbf{A}$ is the adjacency matrix and $\mathbf{d}$ is the degree vector. Tab. 3 exhibit which choice of $\{m^{(i)}\}_{i=1}^4$ correspond to which graph Laplacian.

A question that may arise is whether the innermost $\mathrm{diag}(\cdot)$ in (7) can be factored out of the outermost one. As shown in the next proposition, it is not possible in general.

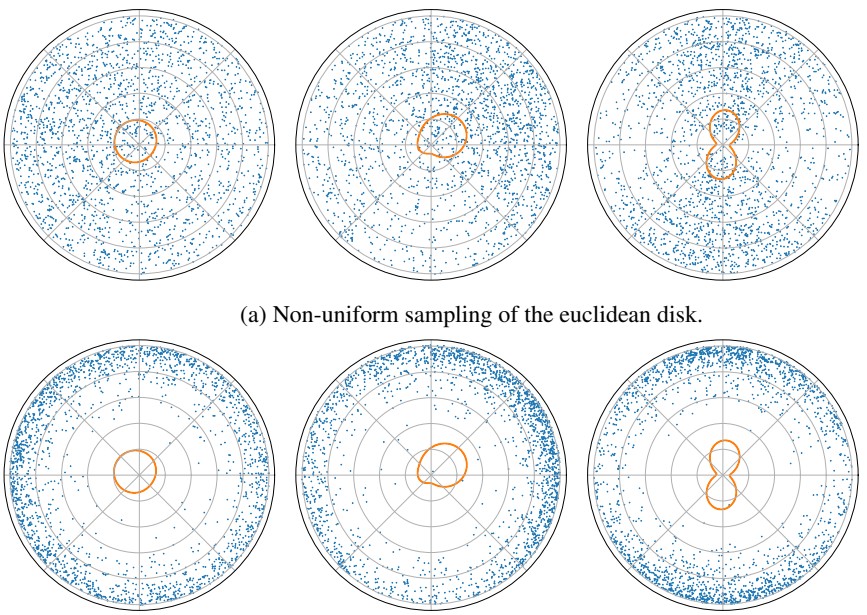

(a) Non-uniform sampling of the euclidean disk.

(b) Non-uniform sampling of the hyperbolic disk.

Figure 9: Examples of non-uniform (a) euclidean and (b) hyperbolic sampling. The orange curve represents the angular probability density function, conveniently rescaled for visibility purposes.

**Proposition 7.** *Let* $\mathbf{A} \in \mathbb{R}^{N \times N}$, $\mathbf{A} = \mathbf{A}^{\mathrm{T}}$*; let* $\mathbf{v} \in \mathbb{R}_{\geq 0}^{N}$ *and* $\mathbf{V} = \mathrm{diag}(\mathbf{v})$*, it holds*

$$\mathrm{diag}(\mathbf{VA1}) = \mathbf{V}\,\mathrm{diag}(\mathbf{A1}) = \mathrm{diag}(\mathbf{A1})\,\mathbf{V}\,.$$

*Moreover*

$$\mathrm{diag}(\mathbf{AV1}) = \mathrm{diag}(\mathbf{VA1}) \iff A_{i,j} = 0 \,\forall\, i,j = 1,\dots,N \,:\, v_i \neq v_j\,.$$

The proof of the statement can be found in Appendix D. An important consequence of Prop. 7 is that the graph Laplacian

$$\mathbf{L} = \mathbf{D}^{-\frac{1}{2}}\mathbf{A}\mathbf{D}^{-\frac{1}{2}} - \mathrm{diag}\left(\mathbf{D}^{-\frac{1}{2}}\mathbf{A}\mathbf{D}^{-\frac{1}{2}}\mathbf{1}\right)\,,\quad \mathbf{D} = \mathrm{diag}(\mathbf{d})\,, \tag{8}$$

obtained with $m^{(i)}(x) = x^{-\frac{1}{2}}$ for every $i \in \{1,\dots,4\}$, is in general different from the symmetric normalized Laplacian, since

$$\mathbf{L} = \mathbf{D}^{-\frac{1}{2}}\mathbf{A}\mathbf{D}^{-\frac{1}{2}} - \mathrm{diag}\left(\mathbf{D}^{-\frac{1}{2}}\mathbf{A}\mathbf{D}^{-\frac{1}{2}}\mathbf{1}\right) \neq \mathbf{D}^{-\frac{1}{2}}\mathbf{A}\mathbf{D}^{-\frac{1}{2}} - \mathbf{D}^{-\frac{1}{2}}\mathrm{diag}(\mathbf{A1})\,\mathbf{D}^{-\frac{1}{2}} = \mathbf{L}_{sn}\,,$$

In light of Prop. 7, the two Laplacians are equivalent if every node is connected to nodes with the same degree, e.g., if the graph is $k$-regular.

The difference between the two Laplacians can be better seen by studying their spectrum. The next proposition introduces an upper bound on the eigenvalues of the Laplacian in (8).

**Proposition 8.** *Let* $\mathcal{G} = (\mathcal{V}, \mathcal{E})$ *be an undirected graph with adjacency matrix* $\mathbf{A} \in \mathbb{R}^{N \times N}$ *and degree matrix* $\mathbf{D} = \mathrm{diag}(\mathbf{A1})$*. Let* $\lambda$ *be an eigenvalue of the graph Laplacian*

$$\mathbf{L} = \mathbf{D}^{-\frac{1}{2}}\mathbf{A}\mathbf{D}^{-\frac{1}{2}} - \mathrm{diag}\left(\mathbf{D}^{-\frac{1}{2}}\mathbf{A}\mathbf{D}^{-\frac{1}{2}}\mathbf{1}\right)\,,$$

*it holds* $|\lambda| \leq 2\sqrt{N}$.

The proof of the proposition can be found in Appendix D. It is well known that the spectral radius of the symmetric normalized Laplacian is less than or equal to 2 (Chung, 1997), with equality holding for bipartite graphs. However, this is not the case for the Laplacian in (8), as shown in the next example.

Table 3: Usual graph shift operators as metric-probability Laplacians.

| Graph Shift Operator | $m^{(1)}(x)$ | $m^{(2)}(x)$ | $m^{(3)}(x)$ | $m^{(4)}(x)$ |
|---|---|---|---|---|
| Adjacency | $1(x)$ | $1(x)$ | $0(x)$ | $0(x)$ |
| Combinatorial Laplacian | $1(x)$ | $1(x)$ | $1(x)$ | $1(x)$ |
| Signless Laplacian (Cvetkovic & Simic, 2009) | $1(x)$ | $1(x)$ | $-1(x)$ | $1(x)$ |
| Random walk Laplacian | $x^{-1}$ | $1(x)$ | $x^{-1}$ | $1(x)$ |
| Right normalized Laplacian | $1(x)$ | $x^{-1}$ | $x^{-1}$ | $1(x)$ |
| Symmetric normalized adjacency (Kipf & Welling, 2017) | $x^{-\frac{1}{2}}$ | $x^{-\frac{1}{2}}$ | $0(x)$ | $0(x)$ |
| Symmetric normalized Laplacian | $x^{-\frac{1}{2}}$ | $x^{-\frac{1}{2}}$ | $x^{-1}$ | $1(x)$ |
| Equation (8) | $x^{-\frac{1}{2}}$ | $x^{-\frac{1}{2}}$ | $x^{-\frac{1}{2}}$ | $x^{-\frac{1}{2}}$ |

**Example 1** (Complete Bipartite Graph). Consider the complete bipartite graph with $n$ nodes in the first part and $m \geq n$ nodes in the second part. Its adjacency and degree matrix are

$$\mathbf{A} = \begin{pmatrix} \mathbf{0}_{n\times n} & \mathbf{1}_{n\times m} \\ \mathbf{1}_{m\times n} & \mathbf{0}_{m\times m} \end{pmatrix} , \ \mathbf{D} = \begin{pmatrix} m\mathbf{I}_{n\times n} & \mathbf{0}_{n\times m} \\ \mathbf{0}_{m\times n} & n\mathbf{I}_{m\times m} \end{pmatrix} .$$

A simple computation leads to

$$\mathbf{L} = \mathbf{D}^{-\frac{1}{2}}\mathbf{A}\mathbf{D}^{-\frac{1}{2}} - \text{diag}\left(\mathbf{D}^{-\frac{1}{2}}\mathbf{A}\mathbf{D}^{-\frac{1}{2}}\mathbf{1}\right) = \begin{pmatrix} -m^{\frac{1}{2}}n^{-\frac{1}{2}}\mathbf{I}_{n\times n} & (nm)^{-\frac{1}{2}}\mathbf{1}_{n\times m} \\ (nm)^{-\frac{1}{2}}\mathbf{1}_{m\times n} & -n^{\frac{1}{2}}m^{-\frac{1}{2}}\mathbf{I}_{m\times m} \end{pmatrix} .$$

It can be noted that $\mathbf{L}$ has null eigenvalue $\lambda_1 = 0$ corresponding to the constant eigenvector $\mathbf{1}_{n+m}$. The vector $\mathbf{v}_i = -\mathbf{e}_1 + \mathbf{e}_i$, $i \in \{2, \ldots, n\}$ is an eigenvector with eigenvalue $\lambda_2 = -\sqrt{m/n}$, whose multiplicity is $n-1$. Analogously, $\mathbf{v}_i = -\mathbf{e}_{n+1} + \mathbf{e}_{i+1}$, $i \in \{n+1, \ldots, n+m-1\}$ is an eigenvector with eigenvalue $\lambda_3 = -\sqrt{n/m}$, whose multiplicity is $m-1$. Finally, the vector $\mathbf{v}_{n+m} = [-m/n\mathbf{1}_n^{\mathrm{T}}, \mathbf{1}_m^{\mathrm{T}}]^{\mathrm{T}}$ is eigenvector with eigenvalues $\lambda_4 = \lambda_2 + \lambda_3$. Therefore, the spectral radius of $\mathbf{L}$ is

$$|\lambda_4| = \frac{m+n}{\sqrt{m\,n}} .$$

In the case of a balanced graph, $n = m$ implies that the spectral radius is 2. In the case of a star graph, $n = 1$ and $|\lambda_4| = \mathcal{O}(\sqrt{m})$ as $m \to \infty$; therefore, the asymptotic in Prop. 8 is tight.

## D  PROOFS

*Proof of Prop. 1 and concentration of error.* Let $\mathbf{x} = \{x_i\}_{i=1}^N$ be an i.i.d. random sample from $\rho$. Let $K$ and $m$ be the kernel and diagonal parts corresponding to the metric-probability Laplacian $\mathcal{L}_\mathcal{N}$. Let $\mathbf{L}, \mathbf{u}$ be

$$L_{i,j} = N^{-1}K(x_i, x_j)\rho(x_j)^{-1} - m(x_i)\,, \ u_i = u(x_i)\,.$$

Note that the non-uniform geometric GSO $\mathbf{L}_{\mathcal{G},\boldsymbol{\rho}}$ based on the graph $\mathcal{G}$, which is randomly sampled from $\mathcal{S}$ with neighborhood model $\mathcal{N}$ via the sample points $\mathbf{x}$, is exactly equal to $\mathbf{L}$. Conditioned on $x_i = x$, the expected value is

$$\mathbb{E}\left(\mathbf{L}\,\mathbf{u}\right)_i = N^{-1}\sum_{j=1}^N \mathbb{E}\left(K(x, x_k)\,\rho(x_j)^{-1}u(x_j)\right) - m(x)u(x) = \mathcal{L}_\mathcal{N}u(x)\,.$$

Since the random variables $\{x_j\}_{j=1}^N$ are i.i.d. to $y$, then also the random variables

$$\left\{K(x, x_j)\rho(x_j)^{-1}\,u(x_j)\right\}_{j=1}^N$$

are i.i.d., hence,

$$\text{var}\left(\mathbf{Lu}\right)_i = \text{var}\left(N^{-1}\sum_{j=1}^N K(x, x_j)\rho(x_j)^{-1}u(x_j) - m(x)\,u(x)\right)$$

$$= N^{-1} \operatorname{var} \left( K(x, y) \rho(y)^{-1} u(y) \right)$$

$$\leq N^{-1} \mathbb{E} \left( \left| K(x, y) \rho(y)^{-1} u(y) \right|^2 \right)$$

$$= N^{-1} \int_{\mathcal{S}} \left| K(x, y) \rho(y)^{-1} u(y) \right|^2 \rho(y) \, \mathrm{d}\mu(y)$$

$$\leq N^{-1} \left\| K(x, \cdot)^2 \rho(\cdot)^{-1} \right\|_{\mathrm{L}^\infty(\mathcal{S})} \|u\|_{\mathrm{L}^2(\mathcal{S})}^2 \,,$$

which proves (5).

Next, we prove the concentration of error result. We know that there exist $a, b > 0$ such that almost everywhere $K(x, x_j) \rho(x_j)^{-1} u(x_j) \in [a, b]$, since $K, 1/\rho$ and $u$ are essentially bounded. By Hoeffding's inequality, for $t > 0$,

$$\mathbb{P} \left[ |(\mathbf{Lu})_i - \mathcal{L}_\mathcal{N} u(x)| \geq t \right] \leq 2 \exp \left( -\frac{2 N t^2}{(b-a)^2} \right) \,.$$

Setting

$$\frac{p}{N} = 2 \exp \left( -\frac{2 N t^2}{(b-a)^2} \right) \,,$$

solving for $t$, we obtain that for every node there is an event with probability at least $1 - p/N$ such that

$$|(\mathbf{Lu})_i - \mathcal{L}_\mathcal{N} u(x)| \leq 2^{-\frac{1}{2}} (b-a) N^{-\frac{1}{2}} \sqrt{\log(2 N p^{-1})} \,.$$

We then intersect all of these events to obtain an event of probability at least $1 - p$ that satisfies (6). □

*Proof of Lemma 1.* By hypothesis, there exist $m_x, M_x > 0$ such that $m_x \leq \rho(y) \leq M_x$ for all $y \in \mathcal{N}(x)$. Therefore,

$$M_x^{-1} \int_{\mathcal{N}(x)} \mathrm{d}\nu(y) \leq \int_{\mathcal{N}(x)} \mathrm{d}\mu(y) = \int_{\mathcal{N}(x)} \rho(y)^{-1} \, \mathrm{d}\nu(y) \leq m_x^{-1} \int_{\mathcal{N}(x)} \mathrm{d}\nu(y) \,,$$

from which

$$m_x \leq \frac{\int_{\mathcal{N}(x)} \mathrm{d}\nu(y)}{\int_{\mathcal{N}(x)} \mathrm{d}\mu(y)} \leq M_x \,.$$

By the Intermediate Value Theorem, there exists $c_x \in \mathcal{N}(x)$ such that

$$\rho(c_x) = \frac{\int_{\mathcal{N}(x)} \mathrm{d}\nu(y)}{\int_{\mathcal{N}(x)} \mathrm{d}\mu(y)} \,,$$

from which the thesis follows. □

*Proof of Prop. 2.* Consider the map

$$\varphi : [-\pi, \pi) \to \mathbb{S}^1 \,, \quad \theta \mapsto \left( \cos(\theta), \quad \sin(\theta) \right)^{\mathrm{T}} \,,$$

and the angles $x, y, \in [-\pi, \pi)$ such that $\varphi(x) = \boldsymbol{x}$, $\varphi(y) = \boldsymbol{y}$, it holds

$$\mathring{d}(\boldsymbol{x}, \boldsymbol{y}) = \arccos(\boldsymbol{x}^{\mathrm{T}} \boldsymbol{y}) = \arccos(\cos(x) \cos(y) + \sin(x) \sin(y))$$

$$= \arccos(\cos(x - y))$$

$$= x - y + 2 k \pi$$

$$= \begin{cases} 2\pi + x - y, & x - y \in [-2\pi, -\pi) \\ y - x, & x - y \in [-\pi, 0) \\ x - y, & x - y \in [0, \pi) \\ 2\pi + y - x, & x - y \in [\pi, 2\pi) \end{cases}$$

$$
= \begin{cases} 2\pi - |x - y|, & |x - y| > \pi \\ |x - y|, & |x - y| < \pi \end{cases}
$$
$$
= \pi - |\pi - |x - y||.
$$

$\square$

*Proof of Prop. 3.* The expected degree of a node $\theta$ is the probability of the ball centered at $\theta$ times the size $N$ of the sample. The probability of a ball can be computed by noting that

$$
\int_{\theta_c - \alpha}^{\theta_c + \alpha} \cos(n_i(\theta - \mu_i)) \, d\theta = \begin{cases} 2\alpha, & n_i = 0 \\ \dfrac{2 \cos(n_i(\theta_c - \mu_i)) \sin(n_i \alpha)}{n_i}, & \text{otherwise} \end{cases}
$$

Therefore, the average degree can be computed as

$$
\overline{d} = N \int_{-\pi}^{\pi} \mathbb{P}[\mathrm{B}_\alpha(\theta)] \, \mathrm{sb}(\theta; \boldsymbol{c}, \boldsymbol{n}, \boldsymbol{\mu}) \, d\theta.
$$

The inspection of $\mathrm{sb}(\theta; \boldsymbol{c}, \boldsymbol{n}, \boldsymbol{\mu})$ and $\mathbb{P}[\mathrm{B}_\alpha(\theta)]$ shows that the only terms surviving integration are the constant term and the product of cosines with the same frequency

$$
\int_{-\pi}^{\pi} \cos(n_i(\theta - \mu_i)) \cos(n_i(\theta - \mu_i)) \, d\theta = \begin{cases} \pi \cos(n_i(\mu_j - \mu_i)), & n_i = n_j \\ 0, & n_i \neq n_j \end{cases}
$$

from which the thesis follows. $\square$

*Proof of Prop. 4.* Using Taylor expansion, it holds

$$
\exp(\kappa_i \cos(n_i(\theta - \mu_i))) = \sum_{m=0}^{\infty} \frac{\kappa_i^m}{m!} \cos(n_i(\theta - \mu_i))^m
$$
$$
= 1 + \sum_{m=1}^{\infty} \frac{\kappa_i^{2m}}{(2m)!} \cos(n_i(\theta - \mu_i))^{2m}
$$
$$
+ \sum_{m=1}^{\infty} \frac{\kappa_i^{2m-1}}{(2m-1)!} \cos(n_i(\theta - \mu_i))^{2m-1}.
$$

A first approximation can be made noting that $\cos(x)^{2m} \leq \cos(x)^2$ and $\cos(x)^{2m-1} \approx \cos(x)$ for all $m \geq 1$, obtaining

$$
\exp(\kappa_i \cos(n_i(\theta - \mu_i))) \approx 1 + (\cosh(\kappa_i) - 1) \cos(n_i(\theta - \mu_i))^2 + \sinh(\kappa_i) \cos(n_i(\theta - \mu_i)).
$$

Such approximation deteriorates fast when $\kappa_i$ increases. A more refined approximation is obtained considering the power of cosine with higher coefficient in the Taylor expansion. Using Stirling's approximation of factorial, it can be shown that

$$
\frac{\kappa_i^m}{m!} \approx \frac{1}{\sqrt{2\pi m}} \left( \frac{\kappa_i e}{m} \right)^m.
$$

In order to make the computation easier, suppose $\kappa_i$ is an integer; When $m = \kappa_i + 1$, it holds

$$
\frac{1}{\sqrt{2\pi(\kappa_i + 1)}} \left( \frac{\kappa_i e}{\kappa_i + 1} \right)^{\kappa_i + 1} = \frac{e^{\kappa_i + 1}}{\sqrt{2\pi(\kappa_i + 1)}} \left( \frac{\kappa_i}{\kappa_i + 1} \right)^{\kappa_i + 1} < \frac{e^{\kappa_i}}{\sqrt{2\pi(\kappa_i + 1)}} < \frac{e^{\kappa_i}}{\sqrt{2\pi \kappa_i}},
$$

where the first inequality is justified by the fact that $(\kappa_i/(\kappa_i + 1))^{\kappa_i + 1}$ is an increasing sequence that tends to $1/e$. The previous formula shows that the coefficient with $m = \kappa_i + 1$ is always smaller than the coefficient with $m = \kappa_i$. The same reasoning can be applied to all the coefficients with

$m > \kappa_i$. Suppose now $\kappa_i \geq 3$, if $m \leq \kappa_i - 2$ the previous reasoning holds. A peculiarity happens when $m = \kappa_i - 1$:

$$\frac{1}{\sqrt{2\pi(\kappa_i - 1)}} \left(\frac{\kappa_i\, e}{\kappa_i - 1}\right)^{\kappa_i - 1} = \frac{e^{\kappa_i - 1}}{\sqrt{2\pi\kappa_i}} \left(\frac{\kappa_i}{\kappa_i - 1}\right)^{\kappa_i - \frac{1}{2}} > \frac{e^{\kappa_i}}{\sqrt{2\pi\kappa_i}},$$

because the sequence $(\kappa_i/(\kappa_i - 1))^{\kappa_i - 1/2}$ is decreasing; therefore $m = \kappa_i - 1$ is the point of maximum, and $m = \kappa_i$ is the second largest value. Therefore, the following approximation for $\exp(\kappa_i \cos(n_i(\theta - \mu_i)))$ holds:

$$\begin{cases} 1 + (\cosh(\kappa_i) - 1)\cos(n_i(\theta - \mu_i))))^2 + \sinh(\kappa_i)\cos(n_i(\theta - \mu_i)))), & \kappa_i \leq 1 \\ 1 + (\cosh(\kappa_i) - 1)\cos(n_i(\theta - \mu_i))))^{\kappa_i} + \sinh(\kappa_i)\cos(n_i(\theta - \mu_i))))^{\kappa_i - 1}, & \kappa_i \geq 1,\ \text{even} \\ 1 + (\cosh(\kappa_i) - 1)\cos(n_i(\theta - \mu_i))))^{\kappa_i - 1} + \sinh(\kappa_i)\cos(n_i(\theta - \mu_i))))^{\kappa_i}, & \kappa_i \geq 1,\ \text{odd} \end{cases}$$

The thesis follows from the equality

$$\cos(n_i(\theta - \mu_i))^{\kappa_i} = \frac{1}{2^{\kappa_i}} \sum_{k=0}^{\kappa_i} \binom{\kappa_i}{k} \cos((2\,k - \kappa_i)n_i(\theta - \mu_i)).$$

$\square$

*Proof of Prop. 5.* The domain of integration can be parametrized as $d_{\mathbb{D}}((r, \theta), (r_c, \theta_c)) \leq \alpha$, leading to

$$\theta \in \left(\theta_c - \arccos\left(\frac{r^2 + r_c^2 - \alpha^2}{2\,r\,r_c}\right), \theta_c + \arccos\left(\frac{r^2 + r_c^2 - \alpha^2}{2\,r\,r_c}\right)\right).$$

Three cases must be discussed: (1) $0 \leq r_c - \alpha \leq r_c + \alpha \leq 1$, (2) $r_c - \alpha < 0$, (3) $r_c + \alpha > 1$. In scenario (1), the ball $\mathrm{B}_\alpha(r_c, \theta_c)$ is contained in $\mathbb{D}$. The probability of the ball can be computed as

$$\begin{aligned} \mathbb{P}\left[\mathrm{B}_\alpha(r_c, \theta_c)\right] &= 2 \int_{r_c - \alpha}^{r_c + \alpha} r \int_{\theta_c - \theta_r}^{\theta_c + \theta_r} \mathrm{sb}(\theta; \boldsymbol{c}, \boldsymbol{n}, \boldsymbol{\mu})\, \mathrm{d}\theta\, \mathrm{d}r \\ &= \frac{4}{B} \sum_{i:n_i \neq 0} \frac{c_i}{n_i} \cos(n_i(\theta_c - \mu_i)) \int_{r_c - \alpha}^{r_c + \alpha} r \sin\left(n_i \arccos\left(\frac{r^2 + r_c^2 - \alpha^2}{2\,r\,r_c}\right)\right) \mathrm{d}r \\ &\quad + \frac{4}{B}\left(\sum_{i:n_i = 0} c_i + A\right) \int_{r_c - \alpha}^{r_c + \alpha} r \arccos\left(\frac{r^2 + r_c^2 - \alpha^2}{2\,r\,r_c}\right) \mathrm{d}r, \end{aligned}$$

(9)

where the last equality comes from Prop. 3. For simplicity, define

$$f_{n_i - 1}(r) = r\sqrt{1 - \left(\frac{r^2 + r_c^2 - \alpha^2}{2\,r\,r_c}\right)^2}\, \mathrm{U}_{n_i - 1}\left(\frac{r^2 + r_c^2 - \alpha^2}{2\,r\,r_c}\right),$$

$$g(r) = r \arccos\left(\frac{r^2 + r_c^2 - \alpha^2}{2\,r\,r_c}\right),$$

where $U_k$ is the $k$-th Chebyshev polynomial of second kind. It is worthy to note that $f_{n_i - 1}(r_c + \alpha) = 0$, $f_{n_i - 1}(r_c - \alpha) = 0$, $f_{n_i - 1}(\alpha - r_c) = 0$ and

$$f_{n_i - 1}(r_c) = \alpha\sqrt{1 - \left(\frac{\alpha}{2\,r_c}\right)^2}\, \mathrm{U}_{n_i - 1}\left(1 - \frac{\alpha^2}{2\,r_c^2}\right), \quad f_{n_i - 1}(\alpha) = \alpha\sqrt{1 - \frac{r_c}{2\,\alpha}}\, \mathrm{U}_{n_i - 1}\left(\frac{r_c}{2\,\alpha}\right),$$

while $g(r_c + \alpha) = 0$, $g(r_c - \alpha) = 0$, $g(\alpha - r_c) = (\alpha - r_c)\pi$ and

$$g(r_c) = r_c \arccos\left(1 - \frac{\alpha^2}{2\,r_c^2}\right), \quad g(\alpha) = \alpha \arccos\left(\frac{r_c}{2\,\alpha}\right).$$

The integral in (9) can be approximated by the semi-area of an ellipse having $\alpha$ and $f_{n_i-1}(r_c)$ (respectively $g(r_c)$) as axes

$$\int\limits_{r_c-\alpha}^{r_c+\alpha} f_{n_i-1}(r)\,\mathrm{d}r \approx \frac{\pi}{2}\alpha f_{n_i-1}(r_c)\,,\quad \int\limits_{r_c-\alpha}^{r_c+\alpha} g(r)\,\mathrm{d}r \approx \frac{\pi}{2}\alpha g(r_c)\,, \tag{10}$$

that can be seen as a modified version of Simpson's rule since the latter would lead to a coefficient of $4/3$ instead of $\pi/2$. A comparison between the two methods is shown in Appendix D. In scenario (2) the domain of integration contains the origin and the argument of $\arccos$ in Appendix D could be not well defined. The singularity can be removed by decomposing the domain of integration as the union of a disk of radius $\alpha - r_c$ around the origin and the remaining. Hence

$$\mathbb{P}\left[\mathrm{B}_\alpha(r_c,\theta_c)\right] = 2\int\limits_0^{\alpha-r_c} r \int\limits_{-\pi}^{\pi} \mathrm{sb}(\theta;\boldsymbol{c},\boldsymbol{n},\boldsymbol{\mu})\,\mathrm{d}\theta\,\mathrm{d}r + 2\int\limits_{\alpha-r_c}^{\alpha+r_c} r \int\limits_{\theta_c-\theta_r}^{\theta_c+\theta_r} \mathrm{sb}(\theta;\boldsymbol{c},\boldsymbol{n},\boldsymbol{\mu})\,\mathrm{d}\theta\,\mathrm{d}r$$

$$= (\alpha - r_c)^2 + 2\int\limits_{\alpha-r_c}^{\alpha+r_c} r \int\limits_{\theta_c-\theta_r}^{\theta_c+\theta_r} \mathrm{sb}(\theta;\boldsymbol{c},\boldsymbol{n},\boldsymbol{\mu})\,\mathrm{d}\theta\,\mathrm{d}r\,.$$

The same reasoning as before leads to the approximations

$$\int\limits_{\alpha-r_c}^{\alpha+r_c} f_{n_i-1}(r)\,\mathrm{d}r \approx \frac{\pi}{2}r_c\,f_{n_i-1}(\alpha)\,,\quad \int\limits_{\alpha-r_c}^{\alpha+r_c} g(r)\,\mathrm{d}r \approx \frac{\pi}{2}r_c\,g(\alpha)\,.$$

In scenario (3) the domain of integration partially lies outside $\mathbb{D}$. Hence

$$\mathbb{P}\left[\mathrm{B}_\alpha(r_c,\theta_c)\right] = 2\int\limits_{r_c-\alpha}^{1} r \int\limits_{\theta_c-\theta_r}^{\theta_c+\theta_r} \mathrm{sb}(\theta;\boldsymbol{c},\boldsymbol{n},\boldsymbol{\mu})\,\mathrm{d}\theta\,\mathrm{d}r\,,$$

that can be approximated as

$$\int\limits_{r_c-\alpha}^{1} f_{n_i-1}(r)\,\mathrm{d}r \approx \frac{f_{n_i-1}(r_c)}{2}\left(\frac{1-r_c}{\alpha}\sqrt{\alpha^2-(1-r_c)^2} + \alpha\,\arcsin\left(\frac{1-r_c}{\alpha}\right)\right)\,,$$

$$\int\limits_{r_c-\alpha}^{1} g(r)\,\mathrm{d}r \approx \frac{g(r_c)}{2}\left(\frac{1-r_c}{\alpha}\sqrt{\alpha^2-(1-r_c)^2} + \alpha\,\arcsin\left(\frac{1-r_c}{\alpha}\right)\right)\,.$$

The three scenarios can be summarized in one big formula. For simplicity, define the operator

$$\mathcal{I}[f](r_c) = \frac{\pi}{2}\frac{\alpha + r_c + \min\{\alpha - r_c, r_c - \alpha\}}{2}f_{n_i-1}\left(\frac{\alpha + r_c + \max\{\alpha - r_c, r_c - \alpha\}}{2}\right)$$

$$- \frac{\max\{r_c+\alpha-1,0\}}{r_c+\alpha-1}\frac{f_{n_i-1}(r_c)}{2}\left(\alpha\,\arccos\left(\frac{1-r_c}{\alpha}\right) - \frac{1-r_c}{\alpha}\sqrt{\alpha^2-(1-r_c)^2}\right)\,,$$

that given a function $f$ returns the ellipse approximation of the integral over balls, it holds

$$\mathbb{P}\left[\mathrm{B}_\alpha(r_c,\theta_c)\right] = \frac{4}{B}\sum_{i:n_i\neq 0}\frac{c_i}{n_i}\cos(n_i(\theta_c-\mu_i))\mathcal{I}[f_{n_i-1}](r_c) + \frac{4}{B}\left(\sum_{i:n_i=0}c_i + A\right)\mathcal{I}[g](r_c)$$

$$+ \max\{0, \alpha - r_c\}^2\,.$$

from which the thesis follows. To compute the average degree of a spatial network from the unit disk, the quantity

$$\bar{d} := \int\limits_0^1 2\,r \int\limits_{-\pi}^{\pi} \mathbb{P}\left[\mathrm{B}_\alpha(r,\theta)\right]\mathrm{sb}(\theta;\boldsymbol{c},\boldsymbol{n},\boldsymbol{\mu})\,\mathrm{d}\theta\,\mathrm{d}r\,,$$

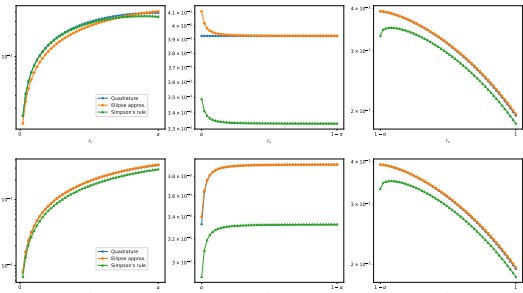

Figure 10: Approximation of $\int_a^b g(r)\,\mathrm{d}r$ (top) and $\int_a^b f_0(r)\,\mathrm{d}r$ (bottom) when $a = \alpha - r_c$ and $b = \alpha + r_c$ (left), $a = r_c - \alpha$ and $b = r_c + \alpha$ (center), and $a = \alpha + r_c$ and $b = 1$ (right) as a function of $r_c$, $\alpha = 0.05$.

must be computed. Using Prop. 3, the integral can be written in the form

$$\bar{d} = \frac{8\pi}{B^2} \sum_{i:n_i \neq 0} \sum_{j:n_i = n_j} \frac{c_i c_j}{n_i} \cos(n_i(\mu_i - \mu_j)) \int_0^1 r\,\mathcal{I}[f_{n_i-1}](r)\,\mathrm{d}r$$

$$+ \frac{16\pi}{B^2} \left( \sum_{i:n_i=0} c_i + A \right)^2 \int_0^1 r\,\mathcal{I}[g](r)\,\mathrm{d}r\,.$$

From $\mathrm{U}_k(1) = k$ the following approximation can be derived

$$\mathcal{I}[f_{n_i-1}](r) \approx \frac{\pi}{2}\,n_i\,\alpha^2 \sqrt{1 - \left(\frac{\alpha}{2\,r}\right)^2} \sim \frac{\pi}{2}\,n_i\,\alpha^2 \left(1 - \frac{\alpha^2}{8\,r^2}\right)\,,$$

hence the integral boils down to

$$\int_0^1 r\,\mathcal{I}[f_{n_i-1}](r)\,\mathrm{d}r \approx \frac{\pi}{4}n_i\alpha^2\,.$$

The term

$$\int_0^1 r\,\mathcal{I}[g](r)\,\mathrm{d}r = \frac{\pi\,\alpha}{24}\left(\alpha\,\sqrt{4-\alpha^2} + 4\,\arccos\left(1 - \frac{\alpha^2}{2}\right)\right)$$

$$+ \frac{\pi\,\alpha^4}{48}\left(\log\left(2 + \sqrt{4-\alpha^2}\right) - \log(\alpha)\right)$$

$$\sim \frac{\pi\,\alpha^2}{24}\left(4 + \sqrt{4-\alpha^2}\right) \sim \frac{\pi\,\alpha^2}{24}\left(6 - \frac{\alpha^2}{4}\right) \sim \frac{\pi\,\alpha^2}{4}\,,$$

from which the thesis follows. $\qquad\square$

*Proof of Prop. 6.* Similarly to what has been done in Prop. 5, the domain of integration can be parametrized as

$$\theta \in (\theta_c - \theta_r, \theta_c + \theta_r)\,,\ \theta_r = \arccos(d_r)\,,\ d_r = \frac{\cosh(r)\cosh(r_c) - \cosh(\alpha)}{\sinh(r_c)\sinh(r)}\,.$$

In order to remove the singularity of the argument of $\arccos$, the domain of integration can be decomposed as a ball containing the origin and the remaining, leading to

$$\mathbb{P}[\mathrm{B}_\alpha(r_c, \theta_c)] = \frac{1}{\cosh(R) - 1} \int_{l_1}^{u_1} \int_{-\pi}^{\pi} \sinh(r)\,\mathrm{sb}(\theta; \boldsymbol{c}, \boldsymbol{n}, \boldsymbol{\mu})\,\mathrm{d}\theta\,\mathrm{d}r$$

$$+ \frac{1}{\cosh(R) - 1} \int_{l_2}^{u_2} \int_{\theta_c - \theta_r}^{\theta_c + \theta_r} \sinh(r)\, \mathrm{sb}(\theta; \boldsymbol{c}, \boldsymbol{n}, \boldsymbol{\mu})\, \mathrm{d}\theta\, \mathrm{d}r\,,$$

where $l_1 = 0$, $u_1 = \max\{\alpha - r_c, 0\}$, $l_2 = |\alpha - r_c|$ and $u_2 = \min\{r_c + \alpha, R\}$.

$$= \frac{\cosh(u_1) - 1}{\cosh(R) - 1} + \frac{1}{\cosh(R) - 1} \int_{l_2}^{u_2} \sinh(r)\, \mathbb{P}\left[\mathrm{B}_{\theta_r}(\theta_c)\right] \mathrm{d}r$$

$$= \frac{\cosh(u_1) - 1}{\cosh(R) - 1} + \frac{2\left(\sum\limits_{i:n_i=0} c_i + A\right)}{B\left(\cosh(R) - 1\right)} \int_{l_2}^{u_2} \sinh(r)\theta_r\, \mathrm{d}r$$

$$+ \frac{2}{B} \sum_{i:n_i \neq 0} \frac{c_i}{n_i} \frac{\cos(n_i(\theta_c - \mu_i))}{\cosh(R) - 1} \int_{l_2}^{u_2} \sinh(r)\sqrt{1 - d_r^2}\, U_{n_i-1}(d_r)\, \mathrm{d}r\,,$$

The approximations $\theta_r \approx \sqrt{2 - 2\,d_r}$ and $d_r \approx 1 + 2\left(e^{-2\,r} + e^{-2\,r_c} - e^{\alpha - r_c - r} - e^{-\alpha - r_c - r}\right)$ as in Gugelmann et al. (2012) can be used to analyze the behaviors of both integrals. For large $R$, it holds

$$\int_{l_2}^{u_2} \frac{\sinh(r)}{\cosh(R) - 1} \theta_r\, \mathrm{d}r \approx \int_{l_2}^{u_2} e^{r-R}\sqrt{1 - d_r}\, \mathrm{d}r$$

$$\approx 2 \int_{l_2}^{u_2} e^{r-R}\sqrt{e^{\alpha - r_c - r} + e^{-\alpha - r_c - r} - e^{-2r} - e^{-2\,r_c}}\, \mathrm{d}r$$

$$= 2 \int_{l_2}^{u_2} e^{\frac{r - 2\,R + \alpha - r_c}{2}} \sqrt{1 + e^{-2\alpha} - e^{-r - \alpha + r_c} - e^{r - \alpha - r_c}}\, \mathrm{d}r$$

$$\approx 4 e^{\frac{\alpha - R - r_c}{2}}\,,$$

where the last approximation is justified by $\sqrt{1 + x} = 1 + \mathcal{O}(x)$ when $|x| \leq 1$. Noting that $-1 \leq d_r \leq 1$, one can get rid of the polynomial contribution

$$\int_{l_2}^{R} \frac{\sinh(r)}{\cosh(R) - 1} \sqrt{1 - d_r^2}\, U_{n_i-1}(d_r)\, \mathrm{d}r \approx n_i \int_{l_2}^{R} e^{r-R}\sqrt{1 - d_r^2}\, \mathrm{d}r$$

$$= n_i \int_{l_2}^{R} e^{r-R}\sqrt{1 - d_r}\sqrt{1 + d_r}\, \mathrm{d}r$$

$$\approx \sqrt{2}\, n_i \int_{l_2}^{R} e^{\frac{r - 2R + \alpha - r_c}{2}} \sqrt{1 + d_r}\, \mathrm{d}r$$

$$\approx 4\, n_i e^{\frac{\alpha - R - r_c}{2}}\,.$$

Therefore, the probability of balls is approximately

$$\mathbb{P}[\mathrm{B}_\alpha(r_c, \theta_c)] \approx \frac{8\, e^{\frac{\alpha - R - r_c}{2}}}{B} \left(\sum_{i:n_i \neq 0} c_i \cos(n_i\,(\theta_c - \mu_i)) + \left(\sum_{i:n_i=0} c_i + A\right)\right)\,,$$

and the expected average degree of the network is

$$\overline{d} = N \int\limits_{0}^{R} \int\limits_{-\pi}^{\pi} \mathbb{P}[\mathrm{B}_{\alpha}(r,\theta)] \, \mathrm{sb}(\theta; \boldsymbol{c}, \boldsymbol{n}, \boldsymbol{\mu}) \frac{\sinh(r)}{\cosh(R) - 1} \, \mathrm{d}\theta \, \mathrm{d}r$$

$$\approx \frac{16 \, \pi \, N \, e^{\frac{\alpha - 2\,R}{2}}}{B^2} \left( \sum_{i:n_i \neq 0} \sum_{j:n_i = n_j} c_i c_j \cos(n_i \, (\mu_i - \mu_j)) + 2 \left( \sum_{i:n_i = 0} c_i + A \right)^2 \right).$$

$\square$

*Proof of Prop. 7.* Equality (2) is trivial, since diagonal matrices commutes; equality (1) follows from

$$\mathrm{diag}(\mathbf{VA1})_{i,i} = (\mathbf{VA1})_i = \sum_{j=1}^{n} V_{i,j}(\mathbf{A1})_j = \sum_{j=1}^{n} \sum_{k=1}^{n} V_{i,j} A_{j,k}$$

$$= \sum_{k=1}^{n} V_{i,i} A_{i,k} = V_{i,i} \sum_{k=1}^{n} A_{i,k} = (\mathbf{V} \, \mathrm{diag}(\mathbf{A1}))_{i,i}.$$

In order to prove (3), we note that $\mathbf{V}$ can be decomposed as $\mathbf{V} = \sum_{i=1}^{n} v_i \, \mathbf{e}^{(i)} \mathbf{e}^{(i)\,\mathrm{T}}$. Therefore

$$0 = (\mathrm{diag}(\mathbf{AV1}) - \mathrm{diag}(\mathbf{VA1}))_{k,k} = \mathrm{diag}\left( \sum_{i=1}^{n} v_i \left( \mathbf{A}\mathbf{e}^{(i)} \mathbf{e}^{(i)\,\mathrm{T}} \mathbf{1} - \mathbf{e}^{(i)} \mathbf{e}^{(i)\,\mathrm{T}} \mathbf{A1} \right) \right)_{k,k}$$

$$= \left( \sum_{i=1}^{n} v_i \left( \mathbf{A}\mathbf{e}^{(i)} - \sum_{j=1}^{n} A_{i,j} \mathbf{e}^{(i)} \right) \right)_k = \sum_{i=1}^{n} v_i A_{k,i} - v_k \sum_{j=1}^{n} A_{k,j}$$

$$= \sum_{i=1}^{n} (v_i - v_k) A_{k,i},$$

must hold for all values of $k$. Consider the indices $k_1, k_2, \ldots, k_n$ corresponding to the values $v_{k_1} \leq v_{k_2} \leq \cdots \leq v_{k_n}$, then

$$0 = \sum_{i=1}^{n} \underbrace{(v_i - v_{k_1})}_{\geq 0} A_{k_1, i},$$

then $A_{k_1, i} = 0$ for each $i$ such that $v_i > v_{k_1}$. Take the index $k_2$ and consider

$$0 = \sum_{i=1}^{n} \underbrace{(v_i - v_{k_2})}_{\geq 0} A_{k_2, i} = \sum_{\substack{i=1 \\ i \neq k_1}}^{n} \underbrace{(v_i - v_{k_2})}_{\geq 0} A_{k_2, i} + \underbrace{(v_{k_1} - v_{k_2}) A_{k_2, k_1}}_{=0}.$$

The second addend is 0 because $v_{k_2}$ can be either equal to $v_{k_1}$, in which case the difference is null, or $v_{k_2} > v_{k_1}$, in which case from the previous step $A_{k_2, k_1} = 0$. Therefore $A_{k_2, i} = 0$ for each $i$ such that $v_i > v_{k_2}$. By finite induction, the thesis holds when $\mathbf{A}$ has null entries in position $(i, j)$ whenever $v_i \neq v_j$. $\square$

*Proof of Prop. 8.* The eigenvalues can be characterized via the Rayleigh quotient

$$\frac{\left\langle \mathbf{u}, \left( \mathrm{diag}\left( \mathbf{D}^{-\frac{1}{2}} \mathbf{A} \mathbf{D}^{-\frac{1}{2}} \mathbf{1} \right) - \mathbf{D}^{-\frac{1}{2}} \mathbf{A} \mathbf{D}^{-\frac{1}{2}} \right) \mathbf{u} \right\rangle}{\langle \mathbf{u}, \mathbf{u} \rangle}.$$

Using Prop. 7, and considering $\mathbf{u} = \mathbf{D}^{\frac{1}{2}} \mathbf{v}$ the previous formula can be rewritten as

$$\frac{\left\langle \mathbf{D}^{\frac{1}{2}} \mathbf{v}, \left( \mathrm{diag}\left( \mathbf{A} \mathbf{D}^{-\frac{1}{2}} \mathbf{1} \right) - \mathbf{D}^{-\frac{1}{2}} \mathbf{A} \right) \mathbf{v} \right\rangle}{\left\langle \mathbf{D}^{\frac{1}{2}} \mathbf{v}, \mathbf{D}^{\frac{1}{2}} \mathbf{v} \right\rangle} = \frac{\mathbf{v}^{\mathrm{T}} \left( \mathrm{diag}\left( \mathbf{D}^{\frac{1}{2}} \mathbf{A} \mathbf{D}^{-\frac{1}{2}} \mathbf{1} \right) - \mathbf{A} \right) \mathbf{v}}{\mathbf{v}^{\mathrm{T}} \mathbf{D} \mathbf{v}}.$$

Let $d_i = \mathbf{D}_{i,i}$ the degree of the $i$-th node, using the symmetry of $\mathbf{A}$, the numerator can be rewritten as

$$\sum_{i,j} v_i^2 \sqrt{\frac{d_i}{d_j}} A_{i,j} - \sum_{i,j} v_i A_{i,j} v_j$$

$$= \frac{1}{2} \sum_{i,j} v_i^2 \sqrt{\frac{d_i}{d_j}} A_{i,j} + \frac{1}{2} \sum_{i,j} v_j^2 \sqrt{\frac{d_j}{d_i}} A_{i,j} - \sum_{i,j} v_i A_{i,j} v_j$$

$$= \frac{1}{2} \left( \sum_{i,j} v_i A_{i,j} \left( \sqrt{\frac{d_i}{d_j}} v_i - v_j \right) - \sum_{i,j} v_j A_{i,j} \left( v_i - \sqrt{\frac{d_j}{d_i}} v_j \right) \right)$$

$$= \frac{1}{2} \sum_{i,j} \left( \frac{v_i}{\sqrt{d_j}} - \frac{v_j}{\sqrt{d_i}} \right) A_{i,j} \left( \sqrt{d_i} v_i - \sqrt{d_j} v_j \right)$$

$$= \frac{1}{2} \sum_{i,j} \frac{A_{i,j}}{\sqrt{d_i d_j}} \left( \sqrt{d_i} v_i - \sqrt{d_j} v_j \right)^2 .$$

From the last equality follows that the eigenvalues are all positive. From $(a - b)^2 \leq 2(a^2 + b^2)$ follows

$$\leq \sum_{i,j} \frac{A_{i,j}}{\sqrt{d_i d_j}} \left( d_i v_i^2 + d_j v_j^2 \right)$$

$$= 2 \sum_{i,j} A_{i,j} \sqrt{\frac{d_i}{d_j}} v_i^2$$

$$\leq 2 \sqrt{N} \sum_i d_i v_i^2$$

$$= 2 \sqrt{N} \, \mathbf{v}^{\mathsf{T}} \mathbf{D} \mathbf{v} ,$$

from which the thesis follows. $\qquad\square$

