# OpenReview forum: "Unveiling the sampling density in non-uniform geometric graphs"
_ICLR.cc/2023/Conference — ICLR 2023 poster_

### Official Review · Reviewer_4pay · 2022-10-24

**Confidence:** 4
**Correctness:** 4
**Technical Novelty And Significance:** 3
**Empirical Novelty And Significance:** 3
**Recommendation:** 8

**Clarity, Quality, Novelty And Reproducibility:**

- The $m$ functions in Defs. 3 and 4 are not defined.
- Related to the above, Def. 4 is hard to parse. What are the interpretations of the different terms composing the Laplacian?
- In Sec. 2.2, the authors claim that "the support of the kernel can be decoupled from the value it attains by defining a suitable neighborhoof model $\mathcal{N}(x)$ [...] and a suitable weighting function $W(x,y)$." However, the relationship between $K(x,y)$, and $\mathcal{N}(x)$ and $W(x,y)$ is not made explicit.
- Figure 5 is hard to read and not properly explained. It might be easier to order the chemical elements by some measure of how rare/discriminative they are.

Minor:

- In the beginning of Sec. 2, the authors claim that "the practitioner has the freedom to design a corresponding graph shift operator". Be careful with such statements---by definition, the graph shift operator should respect the graph structure, i.e., it has to be such that $S_{ij}\neq0$ iff either $(i,j)$ is an edge or $i=j$. Also, it seems like in (2) the GSO is restricted to be the Laplacian?
- Please number important equations.
- Sec. 1 and parts of Sec. 2 are somewhat verbose and repetitive. Some of the text can be cut down to add more experiments.
- Many important experimental details are left to appendices. Consider moving them to the main body of the paper.
- There are a number of typos here and there. Please proofread.
- A limitation of the model in Def. 2 is that it only encompasses graphs which are symmetric. This should be mentioned in the paper.

**Details Of Ethics Concerns:**

N/A.

**Strength And Weaknesses:**

Strengths:

- The problem addressed in this paper is important: indeed, commonly used graph shift operators do not provide precise information about sampling, e.g., it is not clear whether a high-degree node in an *unweighted* graph has high degree because the area surrounding it is densely sampled, or because the sampling radius is large.
- There is a strong motivation to "correct" the graph shift operator by encoding this sampling information: empirically, degree normalized graph shift operators tend to do better than their unnormalized counterparts, so normalizing by an even more accurate measure of sampling density should fare well.
- The numerical results are convincing (though limited).

Weaknesses:

- It is not clear if this framework only applies to unweighted or to both unweighted and weighted graphs. This is an important distinction because while in unweighted graphs it is not possible to discern whether high-degree nodes stem from high sampling density or large neighborhood radius, in weighted graphs the sampling density is more related to the number of edges incident to each node, and the neighborhood radius to their edge weights. Related to this, the paper does not include any numerical results on weighted graphs.
- The "geometric graph with hubs" model is not motivated by real-world networks, but instead, by the ability to decouple the contribution of the sampling density and the neighborhood radius. While the model works well empirically on the synthetic/citation networks considered in the experiments, there is no discussion on whether its assumptions, in particular the slowly varying density assumption, are realistic. For example, it would be important to understand whether canonical graph models typically used to model real-world graphs---such as small-world, preferential attachment, household, etc.---can be modeled as geometric graphs with hubs.
- The numerical experiments are lacking:
   - The link prediction experiments are arguably the most important experiments, as they demonstrate the validity of the model. However, they are only performed on synthetic graphs and on relatively simple (low rank) citation networks, which are embedded in $\mathbb{R}^2$. It would be interesting to see link prediction experiments on more complex networks (perhaps the AIDS dataset?), and with larger embedding dimension.
   - The assumption motivating the self-supervised learning approach---that the task depends mostly on the underlying continuous model---is not rigorously justified, and may be too strong, e.g., in heterophilous graphs.
    - By definition, geometric graphs with hubs have more degrees of freedom than the correlation (inner product) and conventional geometric graphs with which they are compared in Fig. 2. If possible, the authors should include comparisons with other graph models with more degrees of freedom.
- The presentation is lacking and certain concepts are not well-defined, see below.




**Summary Of The Paper:**

This paper introduces a framework allowing to correct the matrix representation (or graph shift operator) of a non-uniform geometric graph according to how this graph is sampled from the underlying metric space, namely, according to a sampling density and a neighborhood radius that is assumed to vary across the graph. Further, it proposes a self-supervised method to estimate the sampling density when the underlying metric space is unknown, and empirically demonstrates that using these corrected graph shift operators improves performance in a number of learning tasks.

**Summary Of The Review:**

The problem addressed in the paper is important and the proposed solution is well-motivated. However, the paper fails to consider weighted graphs; does not rigorously compare the proposed graph model with other models for real-world graphs; and has unclear definitions and limited numerical experiments.

---

> ### Author Response · Authors · 2022-11-18
> **Response to Reviewer 4pay (1/4)**
>
> We are very grateful to the reviewer for the time taken to carefully assess our work and for the valuable feedback. We address each point individually. “Q” quotes the Reviewer and “A” marks the response by the authors.
>
> >**Q. Weighted Graphs**. It is not clear if this framework only applies to unweighted or to both unweighted and weighted graphs. This is an important distinction because while in unweighted graphs it is not possible to discern whether high-degree nodes stem from high sampling density or large neighborhood radius, in weighted graphs the sampling density is more related to the number of edges incident to each node, and the neighborhood radius to their edge weights. Related to this, the paper does not include any numerical results on weighted graphs.
>
> **A**. We focus in this paper on unweighted graphs, but the weighted graph case can be a natural and interesting extension in future work.
> The problem of decoupling the density from the radius is present also for weighted graphs. If there is a cluster of well-connected points, we do not know if they are connected since they were sampled from an area of high density, or since they were sampled from an area of high radius, or both. The weights do not give enough information to determine if we are in either case, unless you define the weight function in a degenerate way, e.g., if the weight is a radial function that decays in a fixed rate independently of the radius.  However, this is not an interesting case.
>
> >**Q. Motivation of geometric graphs with hubs**. The "geometric graph with hubs" model is not motivated by real-world networks, but instead, by the ability to decouple the contribution of the sampling density and the neighborhood radius. While the model works well empirically on the synthetic/citation networks considered in the experiments, there is no discussion on whether its assumptions, in particular the slowly varying density assumption, are realistic.
>
> **A**. We completely agree with this point. In the revised paper, we added the following motivation (Section 2.4, p. 6) which was missing in the previous version:
> “Geometric graphs with hubs are also reasonable from a modeling point of view. For example, it is reasonable to assume that different demographics join a social media platform at different rates. Since the demographic is directly related to the node features, and the graph roughly exhibits homophily, the features are slowly varying over the graph, and hence, so is the sampling density. On the other hand, hubs in social networks are associated with influencers. The conditions that make a certain user an influencer are not directly related to the features. Indeed, if the node features in a social network are user interests, users that follow an influencer tend to share their features with the influencer, so the features themselves are not enough to determine if a node is deemed to be a center of a hub or not. Hence, the radius does not tend to be continuous over the graph, and, instead, is roughly constant and small over most of the graph (non-influencers), except for a number of narrow and sharp peaks (influencers).”
>
> >**Q. Relation between geometric graphs with hubs and other models**. For example, it would be important to understand whether canonical graph models typically used to model real-world graphs---such as small-world, preferential attachment, household, etc.---can be modeled as geometric graphs with hubs.
>
> **A**. Since in the revised paper we both give a heuristic motivation for the model, and we also support the model with extensive experiments (we included additional experiments in the revised paper), we hope the reviewer will agree that this is sufficient motivation and validation of our model for the first paper on this topic. We will leave the theoretical question of whether our model can implement other random graph models as special cases for future work, which is indeed an interesting question.
>
> >**Q. Link prediction experiments**. The link prediction experiments are arguably the most important experiments, as they demonstrate the validity of the model. However, they are only performed on synthetic graphs and on relatively simple (low rank) citation networks, which are embedded in R^2.  It would be interesting to see link prediction experiments on more complex networks (perhaps the AIDS dataset?), and with larger embedding dimension.
>
> **A**. Thank you for this suggestion. We extended the experimental part of the paper and performed the link prediction task on three additional datasets: Amazon Computers, Amazon Photo, and FacebookPagePage (whose statistics, such as the number of nodes and edges, are reported in Tab 1, Appendix A.3, p. 15). Fig. 2 (Section 3.1, p. 7) and Fig. 7 (Appendix A.3, p. 15) show that the NuG model can represent these more complex networks and have accurate and consistent performances across datasets.

---

> > ### Author Response · Authors · 2022-11-18
> > **Response to Reviewer 4pay (2/4)**
> >
> > >**Q. Homophily**. The assumption motivating the self-supervised learning approach---that the task depends mostly on the underlying continuous model---is not rigorously justified, and may be too strong, e.g., in heterophilous graphs.
> >
> > **A**. We apologize that our initial wording was too strong. We weakened our claim in Section 2.5, p. 6 as follows:
> >
> > “The idea behind the proposed method is that the task depends mostly on the underlying continuous model. For example, in shape classification, the label of each graph depends on the surface from which the graph is sampled, rather than the specific intricate structure of the discretization. Therefore, the task network $\Psi$ can perform well if it learns to ignore the particular fine details of the discretization, and  focus on the underlying space. The correction of the GSO via the estimated sampling density (4) gives the network exactly such power. Therefore, we conjecture that $\Theta$ will indeed learn how to estimate the sampling density **for graphs that exhibit homophily**. “
> >
> > >**Q. Comparisons with models with more degrees of freedom**. By definition, geometric graphs with hubs have more degrees of freedom than the correlation (inner product) and conventional geometric graphs with which they are compared in Fig. 2. If possible, the authors should include comparisons with other graph models with more degrees of freedom.
> >
> > **A**. We added a comparison against an auto-encoder with a MLP decoder. We chose this new autoencoder to have roughly the same number of parameters as our geometric graph with hubs auto-encoder, for fair comparison (see Tab. 1, Appendix A.3, p. 15). Fig. 2 (Section 3.1, p. 7) and Fig. 7 (Appendix A.3, p. 15) show that our geometric graph with hubs autoencoder is more accurate. Apart from accuracy, the main point is that our auto-encoder is also more interpretable, which allows us to use it for extracting knowledge from graphs (see Section 3.4 pp. 8-9).
> > To be transparent about the number of parameters, we added Tab. 1 which reports the number of parameters of the graph auto-encoder for each real-world network. We chose to fix the architecture of the encoder, making it very expressive, and choose less expressive decoders. While the total number of parameters is not identical for each of the autoencoders, we believe that our approach helps to isolate the effect of the type of the decoder. Otherwise, if the encoder would be different for different methods (to balance the total number of parameters), it would be hard to discern the different effects of the encoder and the decoder. Note that in all methods the encoder is much more expressive than the decoder.
> >
> > >**Q. Overall Presentation**. The presentation is lacking and certain concepts are not well-defined, see below.
> >
> > **A**.
> > - We now recall in Section 2.1 p. 3 the definition of GSOs, explaining why they are important in graph signal processing and how they are used to define spectral graph convolutional neural networks.
> >
> > - We better explain how to compute an non-uniform geometric GSO without directly sampling the continuous Laplacian, which is unknown.
> >
> > - For readability purposes, we simplified the notation of Defs 3-4 (Section 2.2, p. 4).
> >
> > - To clarify Defs. 3-4, on p.4 we added an example of metric-probability Laplacian deriving from (3) and on p. 5 we reported the corresponding discrete version deriving from (4).
> >
> > - We now explain the significance of Prop. 1 (Section 2.2, p.5).
> >
> > - We give motivation to our geometric graph with hubs model (Section 2.4, p. 6).
> >
> > - We added to Appendix A.3, pp. 14-15 the implementation of all decoders used in the experiments on link prediction.
> >
> > >**Q. Definition of $m^{(i)$**. The $m$ functions in Defs. 3 and 4 are not defined.
> >
> > **A**. We thank the reviewer for spotting it. In Def. 3 (Section 2.2, p. 4) we added:
> > “Let $m^{(i)}:\mathbb{R}\rightarrow\mathbb{R}$ be a continuous function for every $i\in\\{1, \dots, 4\\}$.”
> >
> > >**Q. Definition 4**. Related to the above, Def. 4 is hard to parse. What are the interpretations of the different terms composing the Laplacian?
> >
> > **A**. In order to make Def. 4 (Section 2.2, p. 4) clearer, we changed the formula of the GSO according to the suggestion of the reviewer. More specifically, we introduced the matrices $\mathbf{D}_{\boldsymbol{\rho}}^{(i)} $ that make (4) easier to grasp.
> >
> > >**Q. Neighborhood model**. In Sec. 2.2, the authors claim that "the support of the kernel can be decoupled from the value it attains by defining a suitable neighborhoof model [...] and a suitable weighting function ." However, the relationship between $K$,  and $\mathcal{N}$ and $W$ is not made explicit.
> >
> > **A**. To streamline the paper, as suggested by the reviewers, we deleted this part, as it was not needed.

---

> > > ### Author Response · Authors · 2022-11-18
> > > **Response to Reviewer 4pay (3/4)**
> > >
> > > >**Q. Figure 5**. Figure 5 is hard to read and not properly explained. It might be easier to order the chemical elements by some measure of how rare/discriminative they are.
> > >
> > > **A**. We improved Fig. 5 (Section 3.4, p. 9) by explaining how we compute the rarity of each chemical element:
> > >
> > > “computed as the number of compounds labeled as active (respectively, inactive) containing that particular element divided by the number of active (respectively, inactive) compounds. This is a measure of rarity. For example, potassium is present in $5$ out of $400$ active compounds, and in $1$ over $1600$ inactive compounds. Hence, it is more rare to find potassium in an inactive compound.”
> > >
> > > We now explain how we ordered the chemical elements:
> > >
> > >  “The chemical elements are ordered according to their mean importance, as explained in Section 3.4.”
> > >
> > > >**Q. Definition of graph shift Operator**. In the beginning of Sec. 2, the authors claim that "the practitioner has the freedom to design a corresponding graph shift operator". Be careful with such statements---by definition, the graph shift operator should respect the graph structure, i.e., it has to be such that $S_{i, j} \neq 0$ iff $(i, j)$ either is an edge or $i=j$.
> > >
> > > **A**. Thank you for this comment. We clarified what we mean in the revised paper by recalling a definition of graph shift operators. In Section 2.1, p. 3, we wrote:
> > > “given a graph $\mathcal{G}=(\mathcal{V}, \mathcal{E})$, a GSO is any matrix $\mathbf{L}\in\mathbb{R}^{\lvert \mathcal{V} \rvert \times \lvert \mathcal{V}  \rvert}$ that respects the connectivity of the graph, i.e., $L_{i, j}=0$ whenever $(i, j)\notin \mathcal{E}$, $i\neq j$ (Mateos et al., 2019).”
> > >
> > > >**Q. Equation (2)**. Also, it seems like in (2) the GSO is restricted to be the Laplacian?
> > >
> > > **A**. Eq. (2) (Section 2.1, p. 4) is general. Def. 1 (Section 2.1, p. 3)  of continuous Laplacians is general since $K$ and $m$ are general. Different choices of $K$ and $m$ lead to different metric probability Laplacians and, thus, different sampled graph shift operators in (2).
> > >
> > > >**Q. Labeled equations**. Please number important equations.
> > >
> > > **A**. In the revised paper, every formula that we refer to is numbered.
> > >
> > > >**Q. Sections 1-2**. Sec. 1 and parts of Sec. 2 are somewhat verbose and repetitive. Some of the text can be cut down to add more experiments.
> > >
> > > **A**. We extensively revised and streamlined the text. The main changes are:
> > > - The former section “Related Work” and part of the former “Our Contribution“ is now in the “Introduction” (Section 1, pp. 1-2). The text was streamlined.
> > > - The current “Our Contribution” has been rewritten to spell out our contribution clearly and concisely.
> > > - The former sections “Non-Uniform Geometric Metric-Probability Laplacian” and “Non-Uniform Geometric Graph Laplacian” have been merged into Section 2.2. The text was streamlined.
> > > - The former Section “Learning the Sampling Density for Geometric Graphs with Hubs” has been split into “Geometric Graphs with Hubs” (Section 2.4, pp. 5-6) and “Learning the Sampling Density” (Section 2.5, p. 6) in order to divide our model from the proposed method.

---

> > > > ### Author Response · Authors · 2022-11-18
> > > > **Response to Reviewer 4pay (4/4)**
> > > >
> > > >
> > > >
> > > > >**Q. Main part and appendices**. Many important experimental details are left to appendices. Consider moving them to the main body of the paper.
> > > >
> > > > **A**. In Section 2.1, p. 3 we now recall the definition of GSOs and how they are used to define spectral graph convolutional neural networks:
> > > >
> > > > “GSOs are used in graph signal processing to define filters, as functions of the GSO of the form $f(\mathbf{L})$, where $f:\mathbb{R}\rightarrow\mathbb{R}$ is, a polynomial (Defferrard et al., 2016) or a rational function (Levie et al., 2019).  The filters operate on graph signals $\mathbf{u}$ by $f(\mathbf{L})\mathbf{u}$. Spectral graph convolutional networks are the class of graph neural networks that implement convolutions as filters. When a spectral graph convolutional network is trained, only the filters $f:\mathbb{R}\rightarrow\mathbb{R}$ are learned. One significant advantage of the spectral approach is that the convolution network is not tied to a specific graph, but can rather be transferred between different graphs of different sizes and topologies.”
> > > >
> > > > We added a prototypical example of metric-probability Laplacian that will guide the reader through the crucial definitions of the paper. In particular, we added right after Def. 3 how (3) leads to an approximation of the Laplace-Beltrami operator (Section 2.2, p. 4)
> > > >
> > > > “In order to give a concrete example, suppose the neighborhood radius $\alpha(x)=\alpha$ is a constant, $m^{(1)}(x)=m^{(3)}(x)=x^{-1}$, and  $m^{(2)}(x)=m^{(4)}(x)=1$, then(3)  gives [..] an approximation of the Laplace-Beltrami operator”
> > > >
> > > >  and how its discretization leads to the random walk Laplacian, right beneath Def. 4 (Section 2.2, p. 5):
> > > >
> > > > “For example, in case of $m^{(1)}(x)=m^{(3)}(x)=x^{-1}$, $m^{(2)}(x)=m^{(4)}(x)=1$,  and uniform sampling $\rho=1$, (4) leads to the random-walk Laplacian”
> > > >
> > > > In Section 3.4, p.9 we added a formula to explain how the differentiable pooling layer works:
> > > >
> > > > “a pooling layer [...]  maps the output of the graph neural network $\Psi$ to one feature of the form $\sum_{j=1}^{\lvert \mathcal{V}\rvert}{\rho_j}^{-1}\Psi(\mathbf{X})_j$, where  $\mathbf{X}$ denotes the node features”
> > > >
> > > > Due to the page limit, we could not include many details in the main body.
> > > >
> > > > >**Q. Typos**. There are a number of typos here and there. Please proofread.
> > > >
> > > > **A**. We apologize for the typos. We carefully read the paper and ran it through a spell and grammar checker to find and correct the different mistakes and grammatical issues.
> > > >
> > > > >**Q. Limitations of Def. 2**. A limitation of the model in Def. 2 is that it only encompasses graphs which are symmetric. This should be mentioned in the paper.
> > > >
> > > > **A**. In the revised paper we added a sentence that makes this clear. In Section 2.2, p. 4. we wrote:
> > > > “Since $y\in\mathcal{N}(x)$ implies $x\in\mathcal{N}(y)$ for all $x$, $y\in\mathcal{S}$, Def. 2 models only symmetric graphs.”
> > > >
> > > > >**Q. Summary Of The Review**. The problem addressed in the paper is important and the proposed solution is well-motivated. However, the paper fails to consider weighted graphs; does not rigorously compare the proposed graph model with other models for real-world graphs; and has unclear definitions and limited numerical experiments.
> > > >
> > > > **A**. We believe that we addressed these issues in the revised paper, as explained above. We hope that the reviewer will consider increasing their score in view of the new version of the paper, which we believe was much improved as a result of incorporating the points that the reviewer raised.

---

> > > > > ### Comment · Reviewer_4pay · 2022-11-21
> > > > > **Thank you**
> > > > >
> > > > > Thank you for answering my questions regarding the motivation for the geometric graph with hubs model and the applicability of this framework to weighted graphs.
> > > > >
> > > > > The authors have extensively revised the paper, improving the explanations and writing in Section 2 and including all of the suggested numerical experiments. I am happy with the revision, and will update my score accordingly.

---

### Official Review · Reviewer_EP8m · 2022-10-25

**Confidence:** 3
**Correctness:** 4
**Technical Novelty And Significance:** 3
**Empirical Novelty And Significance:** Not applicable
**Recommendation:** 8

**Clarity, Quality, Novelty And Reproducibility:**

Minor remarks:
- why is the "related work" section at the very end of the paper ? it provides some nice context for the whole paper.
- above Definition 4, you do not use the introduced notation for $q(u)$.
- I haven't found the proof for Lemma 1 in the appendix

**Strength And Weaknesses:**

The problem of non-uniform sampling for geometric graphs is quite natural: it's normal to expect that the sampling measure does not exactly match the density of the graph. The methods developed are sound, and seem to be vindicated by the experiments, although it is interesting that adding density estimation mainly reduces the variance and does not affect too much the average/best performance.

On the other hand, the paper is a bit too technical in several aspects, which might hinder its readability for people that are not at the exact field intersection.
- on the theoretical side, it glosses quite quickly over definitions 3 and 4. In particular, the functions m^{(i)} are added without any explanation, except the one given in Appendix C; it might be more illuminating to move part of the appendix to the main text, so that readers used to classical graph theory can make the link between the usual Laplacians and the one in Equation 3. I also don't see the point of introducing the multiplication operator for Definition 3, as opposed to the more explicit forms used previously. It does make a nice parallelism between $\mathcal M$ and the diagonal matrices of Definition 4, but if that is the goal it should be made more explicit.
- the experiment section makes several references to the internals of laplacian-based networks, especially the pooling layers; it would be interesting to see how the density estimation exactly factors into that, even as an additional appendix.

**Summary Of The Paper:**

This paper is interested in the study of geometric graphs, i.e. graphs sampled from an underlying metric space, such that the neighbourhoods are determined by the graph distance. In particular, it focuses on the case where the sampling measure $\nu$ differs from the intrisic measure $\mu$ of the underlying space (i.e. the one used to compute the classical Laplacian).

The authors develop a general form for a neighbourhood function and a geometric Laplacian, which can be understood as a Laplacian whose kernel $K(x, \cdot)$ is simply the indicator of a given ball around $x$. They also provide the corresponding definition for the discrete graph Laplacian, which depends on the sampling density $\rho = d\nu/d\mu$.

The next step is thus to estimate $\rho$; the authors introduce the concept of "Geometric graph with Hubs", which approximates real-worlds graphs decently well. In this model, $\rho$ is approximately proportional to the graph degree, and hence $\rho^{-1}$ can be learned by an equivariant neural network taking as input the node degrees.

Finally, they provide several numerical experiments showing how the density estimation step improves the performance of classical graph convolutional networks. They compare the performance when the Laplacian is corrected with $\rho$ and when it's not, as well as when the correction is only done in the last pooling layer.

**Summary Of The Review:**

This is a good paper, that could use some more work to broaden its appeal to non-expert readers.

---

> ### Author Response · Authors · 2022-11-18
> **Response to Reviewer EP8m (1/2)**
>
> We are very grateful to the reviewer for the time taken to carefully assess our work and for the valuable feedback. We address each point individually. “Q” quotes the Reviewer and “A” marks the response by the authors.
>
> >**Q. Technical details**. On the other hand, the paper is a bit too technical in several aspects, which might hinder its readability for people that are not at the exact field intersection.
>
> **A**. In order to make the background clearer, we now recall in Section 2.1 p. 3 the definition of GSOs, explaining why they are important in graph signal processing and how they are used to define spectral graph convolutional neural networks. We now explain in Section 3.4, p. 9  how  our differentiable pooling is performed.
>
> For readability purposes, we simplified the notation of Defs 3-4 (Section 2.2, p. 4).
>
> To clarify Defs. 3-4, on p.4 we added an example of metric-probability Laplacian deriving from (3) and on p. 5 we reported the corresponding discrete version deriving from (4).
>
> We now explain the significance of Prop. 1 (Section 2.2, p. 5)
>
> We added to Appendix A.3, pp. 14-15 the implementation of all decoders used in the experiments on link prediction.
>
> >**Q. Technical details for the theory**. on the theoretical side, it glosses quite quickly over definitions 3 and 4. In particular, the functions m^{(i)} are added without any explanation, except the one given in Appendix C; it might be more illuminating to move part of the appendix to the main text, so that readers used to classical graph theory can make the link between the usual Laplacians and the one in Equation 3.
>
> **A**. Thank you for pointing out the deficiencies in our exposition. We changed the paper as follows. In Def. 3 (Section 2.2, p. 4)  we added:
>
> “Let $m^{(i)}:\mathbb{R}\rightarrow\mathbb{R}$ be a continuous function for every $i\in\\{1, \dots, 4\\}$.”
>
> We added right after Def. 3 how (3) leads to an approximation of the Laplace-Beltrami operator (Section 2.2, p. 4):
>
> “In order to give a concrete example, suppose the neighborhood radius $\alpha(x)=\alpha$ is a constant, $m^{(1)}(x)=m^{(3)}(x)=x^{-1}$, and  $m^{(2)}(x)=m^{(4)}(x)=1$, then (3) gives [...]  an approximation of the Laplace-Beltrami operator.”
>
>  We also wrote how the discretization of this Laplacian leads to the random walk Laplacian, right beneath Def. 4 (Section 2.2, p. 5):
>
> “For example, in case of $m^{(1)}(x)=m^{(3)}(x)=x^{-1}$, $m^{(2)}(x)=m^{(4)}(x)=1$,  and uniform sampling $\rho=1$, (4) leads to the random-walk Laplacian”
>
> Due to the page limit, we were not able to include more from Appendix C.
>
> >**Q. Formulation of Definition 3**. I also don't see the point of introducing the multiplication operator for Definition 3, as opposed to the more explicit forms used previously. It does make a nice parallelism between $\mathcal{M}$ and the diagonal matrices of Definition 4, but if that is the goal it should be made more explicit.
>
> **A**. We agree that Defs. 3-4 (Section 2.2, p. 4) were too technical in the original version of the paper. Therefore, we changed them to improve readability and clarity. Specifically, we removed the operator notation in Def. 3 making explicit use of integrals instead. In Def. 4, we simplify the notation introducing the matrices $\mathbf{D}_{\boldsymbol{\rho}}^{(i)}$.
>
> >**Q. Details of the density-corrected spectral methods**. the experiment section makes several references to the internals of laplacian-based networks, especially the pooling layers; it would be interesting to see how the density estimation exactly factors into that, even as an additional appendix.
>
> **A**. Thank you for this important comment. We revised the paper accordingly, which improved its readability. In Section 2.1, p. 3 we now recall the definition of GSO:
>
> “ given a graph $\mathcal{G}=(\mathcal{V}, \mathcal{E})$, a GSO is any matrix $\mathbf{L}\in\mathbb{R}^{\lvert \mathcal{V} \rvert \times \lvert \mathcal{V}  \rvert}$ that respects the connectivity of the graph, i.e., $L_{i, j}=0$ whenever $(i, j)\notin \mathcal{E}$, $i\neq j$”
>
> and right after we recall how GSO are implemented in spectral graph convolutional networks:
>
> “GSOs are used in graph signal processing to define filters [...] Spectral graph convolutional networks are the class of graph neural networks that implement convolutions as filters.”
> In Section 3.4, p.9 we added a formula to explain how the differentiable pooling layer works:
> “a pooling layer [...]  maps the output of the graph neural network $\Psi$ to one feature of the form $\sum_{j=1}^{\lvert \mathcal{V}\rvert}{\rho_j}^{-1}\Psi(\mathbf{X})_j$, where  $\mathbf{X}$ denotes the node features”
>
> We added to Appendix A.3, pp. 14-15 the implementation of all decoders used in the experiments on link prediction. We added Tab. 1 which displays the statistics of the real-word networks used for link prediction and node classificatios, as well as the number of parameters of their corresponding graph auto-encoder.

---

> > ### Author Response · Authors · 2022-11-18
> > **Response to Reviewer EP8m (2/2)**
> >
> > >**Q. Related work section**. why is the "related work" section at the very end of the paper? it provides some nice context for the whole paper.
> >
> > **A**. We merged the “Related Work” with the “Introduction” (Section 1, pp. 1-2) since, as the reviewer rightly pointed out, it gives context for the development of the theory.
> >
> > >**Q. Notation for entry-wise functions**. above Definition 4, you do not use the introduced notation for q(u)
> >
> > **A**. We used the notation $q(\mathbf{u})$ when defining the matrices $\mathbf{D}_{\boldsymbol{\rho}}^{(i)}$ in Def. 4 (Section 2.3, p. 4 of the old version). To clarify the connection, we changed the notation $q$ to $m$, so that the link with the functions $m^{(i)}$ in Def. 4 is more visible (Section 2.3, p. 4 of the new version).
> >
> > >**Q. Lemma 1**. I haven't found the proof for Lemma 1 in the appendix
> >
> > **A**. We added the proof of Lemma 1 to Appendix D, p. 21.

---

> ### Author Response · Authors · 2022-12-12
> **Feedback on the revised paper**
>
> Dear Reviewer EP8m,
>
> thank you again very much for your careful reading of our manuscript. Since the Stage 2 deadline is approaching, this is a gentle reminder to acknowledge our revision.

---

> > ### Comment · Reviewer_EP8m · 2022-12-12
> > **Acknowledgement**
> >
> > Dear authors,
> > Thank you for the reminder ! As the clarity of the paper has been substantially improved, I am happy to raise my score.
> > One last remark : in Eq. (3), the terms depending only on x should be outside of the integral.

---

### Official Review · Reviewer_Gd1j · 2022-11-02

**Confidence:** 2
**Clarity, Quality, Novelty And Reproducibility:** The writing is not good.
**Correctness:** 3
**Technical Novelty And Significance:** 2
**Empirical Novelty And Significance:** 2
**Recommendation:** 5

**Strength And Weaknesses:**

Strength: The topic seems quite interesting and is of high practical value.

Weakness: the presentation of the paper overall is not good. There are lots of typos and grammar issues that prevent me from understanding the paper well. Authors are suggested to overhaul the whole organization of the paper as well.

1) Subsection 1.1 on page 2, titled *Our Contributions* is poorly written. It actually states not *contributions* but more about *motivations*. Thus, I suggest integrating it into part of the introduction.

2) Hard to appreciate the contributions in the paper. Suggestions: please state the primary approach to estimate the unknown sampling density, highlight the differences between the proposed method and existing ones, and identify challenges that arise from uniform sampling to non-uniform sampling. Especially please explain in detail the implications and significance of Proposition 1 on top of page 5 in the estimation of the sampling density. I was given the impression that the techniques presented in the paper are just simple generalizations of the existing ones (perhaps I am wrong).

**Summary Of The Paper:**

The paper has considered non-uniformly sampled graphs from a metric-probability space and developed methods to estimate the unknown sampling density from those graphs. Also, the authors have experimentally tested the model and approach on synthetic and real-world graphs.

**Summary Of The Review:**

See the above.

---

> ### Author Response · Authors · 2022-11-18
> **Response to Reviewer Gd1j (1/2)**
>
> We are very grateful to the reviewer for the time taken to carefully assess our work and for the valuable feedback. We address each point individually. “Q” quotes the Reviewer and “A” marks the response by the authors.
> >**Q. General weakness**. the presentation of the paper overall is not good. There are lots of typos and grammar issues that prevent me from understanding the paper well. Authors are suggested to overhaul the whole organization of the paper as well.
>
> **A**. We are sorry for the typos and grammatical errors. We extensively revised and streamlined the paper and corrected (hopefully) all of the grammatical errors.
> We also revised the organization of the paper.
> - The former section “Related Work” and part of the former “Our Contribution“ is now in the “Introduction” (Section 1, pp. 1-2), which we shortened in order to streamline the paper.
> - The “Our Contribution” (Section 1.1, pp. 2-3) has been rewritten to clarify the novelties introduced by our paper. Specifically, paragraph 1 deals with the difference between our and traditional approaches, paragraph 2 explains the difference between the NuG model and the standard geometric graph model, and paragraph 3 introduces the idea behind our novel method to learn the underlying sampling density.
> - The former sections “Non-Uniform Geometric Metric-Probability Laplacian” and “Non-Uniform Geometric Graph Laplacian” have been merged into Section 2.2, pp. 4-5 to streamline the paper and give a more coherent development of the theory.
> - The former Section “Learning the Sampling Density for Geometric Graphs with Hubs” has been split into “Geometric Graphs with Hubs” (Section 2.4, pp. 5-6 ) and “Learning the Sampling Density” (Section 2.5, p. 6) to divide our model of geometric graphs with hubs from the proposed method to learn the underlying sampling density.
>
> >**Q. Contribution section**. Subsection 1.1 on page 2, titled Our Contributions is poorly written. It actually states not contributions but more about motivations. Thus, I suggest integrating it into part of the introduction.
>
> **A**. We re-wrote the section “Our Contributions” (Section 1.1, pp. 2-3) to list the contributions of the paper clearly. We structured it to highlight
> - the main difference between our approach and traditional ones (paragraph 1);
> - the differences between our NuG model and the standard geometric graph model (paragraph 2);
> - the idea behind our proposed method for learning the underlying sampling density in a self-supervised manner
>
> >**Q. Clarity of exposition**. Hard to appreciate the contributions in the paper. Suggestions: please state the primary approach to estimate the unknown sampling density, highlight the differences between the proposed method and existing ones, and identify challenges that arise from uniform sampling to non-uniform sampling.
>
> **A**. We added text to the “Our Contribution” section that explains challenges that arise from having non-uniform density and neighborhood radii (Section 1.1, p. 2, paragraph 3):
>
> “Estimating these by only observing the graph is a hard task. For example, graph quantities like the node degrees are affected both by the density and the radius, and hence, it is hard to decouple the density from the radius by only observing the graph.”
>
> We then added text to “Our Contribution” that states the primary approach to estimate the unknown sampling density:
>
> “We hence propose methods for estimating the density (and radius) using a self-supervision approach. The idea is to train, against some arbitrary task, a spectral graph neural network, where the GSOs underlying the convolution operators are taken as a non-uniform geometric GSO with learnable density. For the model to perform well, it learns to estimate the underlying sampling density, even though it is not directly supervised to do so.”
>
> We also wrote a new section (2.5 Learning the Sampling Density, p. 6) to clearly state the method for estimating the density. This section gives motivation and details on how self-supervision is done.
>
> Lastly, the full implementation details are given in Appendix A.2.
>
> Regarding the last request, to the best of our knowledge, there are no density estimation methods similar to the one proposed in our paper. What is usually done in the literature is the opposite: a metric-probability space and a sampling density are chosen, and the validity of the model is verified experimentally. We are interested in the inverse problem - estimating the density and radius from the observed graph - which has much more practical importance.

---

> > ### Author Response · Authors · 2022-11-18
> > **Response to Reviewer Gd1j (2/2)**
> >
> > >**Q. Significance of Proposition 1**. Especially please explain in detail the implications and significance of Proposition 1 on top of page 5 in the estimation of the sampling density.
> >
> > **A**. The point in this proposition is that if we know the sampling density and the graph structure, we can derive a GSO that approximates the metric-probability Laplacian model, without explicitly knowing the latter. This does not solve the problem of estimating the density, which is solved in the subsequent sections. Since this was not clear in the original version of the paper, we revised the paper as follows.
> > We first added the following text to make clear that as a modeling assumption, we suppose that graphs are sampled from underlying metric structures (p. 4, paragraph before Def. 4):
> >
> > “Since the neighborhood model of $\mathcal{S}$ represents adjacency in the metric space, we make the modeling assumption that graphs are sampled from neighborhood models, as follows….”
> >
> > We then explain how this assumption allows computing a GSO that sampled the continuous Laplacian, without knowing the continuous Laplacian:
> >
> > “Now, a GSO can be sampled from a metric-probability Laplacian model   $\mathcal{L}\_\mathcal{N}$ by (2), if the underlying continuous model is known. However, such knowledge is not required, since the special structure of the metric-probability Laplacian model allows deriving the GSO directly from the sampled graph $\mathcal{G}$ and the sampled density $\\{\rho(x_i)\\}_{i=1}^N$.”
> >
> > We then added the following text before Prop. 1 (p. 5):
> > “The non-uniform geometric GSO in Def. 4 is the Monte-Carlo approximation of the metric-probability Laplacian in Def. 3. This is shown in the following proposition, whose proof can be found in Appendix D.”
> >
> > and after the proposition wrote:
> >
> > “Prop. 1 means that if we are given a graph that was sampled from a neighborhood model, and we know (or have an estimate of) the sampling density at every node of the graph, then we can compute a GSO according to (4) that is guaranteed to approximate a corresponding unknown metric-probability Laplacian. The next goal is hence to estimate the sampling density from a given graph.”
> >
> > >**Q. Significance**. I was given the impression that the techniques presented in the paper are just simple generalizations of the existing ones (perhaps I am wrong).
> >
> > **A**. The Monte-Carlo analysis is indeed rather standard, but this is not the contribution of our paper. Conventional approaches for approximating continuous Lapalcians with graph Laplacians consider the direct approach: using the known continuous Laplacian, we can compute a graph Laplacian that approximates it, proving the approximation with Monte-Carlo theory. We consider the inverse problem, which is the novelty in our paper. We show that if you only observe the graph, and if you have an estimate of the density, you can construct a graph Laplacian that is guaranteed to approximate a latent continuous Laplacian. We then also explain that estimating the density is reasonable, and support our claims with a thorough experimental investigation.
> >
> > To make this key point of our paper clear, we added the following sentence to the “Our Contribution” section (Section 1.1, p. 2, paragraph 1):
> >
> > “While traditional Laplacian approximation approaches solve the direct problem -- approximating a known continuous Laplacian with a graph Laplacian -- in this paper we solve the inverse problem -- constructing a graph Laplacian from an observed graph that is guaranteed to approximate an unknown continuous Laplacian. We believe that our approach has high practical significance, as in practical data science on graphs, the graph is typically given, but the underlying continuous model is unknown.”
> >
> > >**Q. Quality of writing**. The writing is not good.
> >
> > **A**. We overhauled the paper and ran it through a spell and grammar checker. We also revised and streamlined the structure of the paper, as explained above.
> >
> > >**Q. Correctness**. 3: Some of the paper’s claims have minor issues. A few statements are not well-supported, or require small changes to be made correct.
> >
> > **A**. The reviewer did not point to any specific error in our theoretical derivation or claims. Each of our theoretical claims is supported by a proof, and the experimental claims are supported by experiments. Although we do not believe that there is an error, we may have missed something, and we would appreciate it if the reviewer could point to a specific error so we can correct it. Otherwise, we would appreciate it if the reviewer raised their score.

---

> ### Author Response · Authors · 2022-12-12
> **Feedback on the revised paper**
>
> Dear Reviewer Gd1j,
>
> thank you again very much for your careful reading of our manuscript. Since the Stage 2 deadline is approaching, this is a gentle reminder to acknowledge our revision.

---

### Official Review · Reviewer_swjz · 2022-11-06

**Confidence:** 3
**Correctness:** 4
**Technical Novelty And Significance:** 3
**Empirical Novelty And Significance:** 2
**Recommendation:** 6

**Clarity, Quality, Novelty And Reproducibility:**

This work in general could be more clearly written, starting from motivating properly what this work is trying to do and why. For example, do we just want to derive a proper normalization of the adjacency matrix in a way that improves classification performance, or we want to explain in a theoretical grounded way why this normalization is necessary? Also, I think section 2 is quite verbose and each subsection does not seamlessly build on what is discussed in the previous one. E.g., section 2.2 starts by discussing again eq 1 with little reference to what was discussed right before (in deriving eq 2).
In terms of quality more could be done in the experimental section with respect to comparing with other methods for learning parameterized graph shift operators, or a discussion with other latent models that try to learn graphs and assume different levels of popularity for each node.
In terms of originality of this work, it studies geometric graphs in a different context than it was done before. So far, geometric graphs were (mostly) assumed to be uniform as they were mostly studied as a theoretical model with concrete expressions on properties of them, e.g. degree distribution. Hence the uniformity assumption to derive such expression. From the GSO side, this work takes a rather different angle, rather than learning the parameters, it learns a way to normalize the adjacency matrix.

**Strength And Weaknesses:**

*  After the introduction of the non-uniform geometric graph model, the resulting GSO is properly derived with a bound on the convergence of it to the continuous laplacian.
* There is an admirable effort to justify the reasoning behind using the proposed geometric model through experiments where the decoder is restricted to be a geometric graph with hubs and with experiments that try to demonstrate the applicability of this work in practical tasks.
* However, as the work builds on the assumption of slowly changing density functions and piece-wise neighborhood radius, there should be a better discussion of it with other models, e.g. in statistical models there is a notion of popularity/expansiveness that differs for each node [1].
* From a clarity perspective, this work is, in my opinion, weakly motivated. For example, the relevant section starts with a discussion on ways to compare graphs for similarity. It is not clear how this is connected with this work. Moreover, while GSO are central to this work, the definition on them is not repeated in order to start a discussion on how they are used and why this approach can be better than other approaches. In general, reading this work leaves you with a question-mark on what the main target of it is.
* Moreover, in the experimental section regarding classification accuracy there is no comparison with works that learn a parameterized form of GSO (like the one of dasoulas et al. cited in this work). It is a question of whether we could have both pgso and this work combined.
* Finally, for the learned $1/\rho$, of the real-world networks, it would be interesting to see some plots and how they are different from let's say $1/\sqrt{deg}$.


[1] https://arxiv.org/pdf/0912.5410.pdf

**Summary Of The Paper:**

This work considers geometric graphs having both non-uniform sampling density, as well varying neighborhood radius. Under this model, a GSO can be though of as a discretization of the latent continuous Laplacian. In order for this GSO to approximate the continuous laplacian, the adjacency matrix needs to be normalized according to the sampling density. The non-uniform geometric graph model considered here seems a plausible model for real-world graphs. Of course, though, the sampling density is not known in practice. For this reason, it estimates the sampling density using a NN in a self-supervised manner. Experiments on synthetic datasets, where the ground truth sampling density is known, show that indeed this NN approach can well approximate the true underlying sampling density. Finally, it concludes with a set of experiments that seem to validate the hypotheses stated earlier in this work. More precisely, performance is improved in classification tasks, while learning density values can be used as a way to assign importance scores to nodes.

**Summary Of The Review:**

I believe this work has some potential, though certain things need to be addressed. Mostly related to clarity, better motivating this work and focussing on the exact question/problem it tries to answer/solve. This will drive both the theoretical explanation and empirical evaluation in a more clear and concrete way.

---

> ### Author Response · Authors · 2022-11-18
> **Response to Reviewer swjz (1/3)**
>
> We are very grateful to the reviewer for the time taken to carefully assess our work and for the valuable feedback. We address each point individually. “Q” quotes the Reviewer and “A” marks the response by the authors.
>
> >**Q. Assumptions on sampling density and neighborhood radius**. However, as the work builds on the assumption of slowly changing density functions and piece-wise neighborhood radius, there should be a better discussion of it with other models, e.g. in statistical models there is a notion of popularity/expansiveness that differs for each node [1].
>
> **A**. Thank you for this important point. In the revised paper, we added the following motivation (Section 2.4, p. 6) which was missing in the previous version:
>
> “Geometric graphs with hubs are also reasonable from a modeling point of view. For example, it is reasonable to assume that different demographics join a social media platform at different rates. Since the demographic is directly related to the node features, and the graph roughly exhibits homophily, the features are slowly varying over the graph, and hence, so is the sampling density. On the other hand, hubs in social networks are associated with influencers. The conditions that make a certain user an influencer are not directly related to the features. Indeed, if the node features in a social network are user interests, users that follow an influencer tend to share their features with the influencer, so the features themselves are not enough to determine if a node is deemed to be a center of a hub or not. Hence, the radius does not tend to be continuous over the graph, and, instead, is roughly constant and small over most of the graph (non-influencers), except for a number of narrow and sharp peaks (influencers).”
>
> >**Q. Clarity & motivations**. From a clarity perspective, this work is, in my opinion, weakly motivated. For example, the relevant section starts with a discussion on ways to compare graphs for similarity. It is not clear how this is connected with this work.
>
> A. We extensively edited and streamlined the paper. We deleted the references on graph similarity. With respect to clarity, we refer to the answer to the last Q.
>
> >**Q. GSO an their use in spectral approaches**. Moreover, while GSO are central to this work, the definition on them is not repeated in order to start a discussion on how they are used and why this approach can be better than other approaches. In general, reading this work leaves you with a question-mark on what the main target of it is.
>
> **A**. We added in the revised paper the definition of GSOs (Section 2.1, p. 3):
>
> “Loosely speaking, given a graph $\mathcal{G}=(\mathcal{V}, \mathcal{E})$, a GSO is any matrix $\mathbf{L}\in\mathbb{R}^{\lvert \mathcal{V} \rvert \times \lvert \mathcal{V}  \rvert}$ that respects the connectivity of the graph, i.e., $L_{i, j}=0$ whenever $(i, j)\notin \mathcal{E}$, $i\neq j$ (Mateos et al., 2019).”
>
> To explain why they are important and how they are used, the sentence continues:
>
> “GSOs are used in graph signal processing to define filters, as functions of the GSO of the form $f(\mathbf{L})$, where $f:\mathbb{R}\rightarrow\mathbb{R}$ is, e.g., a polynomial (Defferrard et al., 2016) or a rational function (Levie et al., 2019). The filters operate on graph signals $\mathbf{u}$ by $f(\mathbf{L})\mathbf{u}$. Spectral graph convolutional networks are the class of graph neural networks that implement convolutions as filters. When a spectral graph convolutional network is trained, only the filters $f:\mathbb{R}\rightarrow\mathbb{R}$ are learned. One significant advantage of the spectral approach is that the convolution network is not tied to a specific graph, but can rather be transferred between different graphs of different sizes and topologies.”
>
> To explain the connection between GSO and sampling density estimation, we extended the text as follows. Section 1.1, p. 2 now explains how GSOs are used to estimate the sampling density:
>
> “We hence propose methods for estimating the density (and radius) using a self-supervision approach. The idea is to train, against some arbitrary task, a spectral graph neural network, where the GSOs underlying the convolution operators are taken as a non-uniform geometric GSO with learnable density. For the model to perform well, it learns to estimate the underlying sampling density, even though it is not directly supervised to do so.”.
>
> Section 2.2, p. 4 now clarifies how graphs are generated:
>
> “We make the modeling assumption that graphs are sampled from neighborhood models, as follows. First,  random independent points $\mathbf{x}=\{x_i\}_{i=1}^N$ are sampled from $\mathcal{S}$ according to the ``non-uniform'' distribution $\nu$ as before. Then, an edge is created between each pair $x_i$ and $x_j$ if $x_j\in\mathcal{N}(x_i)$, to form the graph $\mathcal{G}$.”

---

> > ### Author Response · Authors · 2022-11-18
> > **Response to Reviewer swjz (2/3)**
> >
> >
> >
> > Then, it clarifies how to compute the non-uniform geometric GSO from the graph structure and the known sample of the density, without explicitly evaluating the continuous Laplacian:
> >
> > “ Now, a GSO can be sampled from a metric-probability Laplacian model   $\mathcal{L}\_{\mathcal{N}}$ by (2) if the underlying continuous model is known. However, such knowledge is not required, since the special structure of the metric-probability Laplacian model allows deriving the GSO directly from the sampled graph $\mathcal{G}$ and the sampled density $\\{\rho(x_i)\\}_{i=1}^N$.”
> >
> > Finally, Section 2.5, p. 6 introduces our method to approximate the sampling density
> >
> > “Since in real-world scenarios the ground-truth density is not known, we train $\Theta$ [the density estimator, A/N] in a self-supervised manner. In this context, we choose a task (link prediction, node or graph classification, etc.) on a real-world graph $\mathcal{G}$ and we solve it by means of a  graph neural network $\Psi$, referred to as \emph{task network}. Since we want  $\Psi$ to depend on the sampling density estimator $\Theta$, we define $\Psi$ as a spectral graph convolution network based on the non-uniform geometric GSO  $\mathbf{L}_{\mathcal{G},\Theta(\mathcal{G})}$”
> >
> > >**Q. Additional experiments**. Moreover, in the experimental section regarding classification accuracy there is no comparison with works that learn a parameterized form of GSO (like the one of dasoulas et al. cited in this work). It is a question of whether we could have both pgso and this work combined.
> >
> > **A**. In the revised paper we compared our method on additional datasets (Amazon Computers, Amazon Photo, FacebookPagePage) and against additional auto-encoders (namely, an auto-encoder with general MLP decoder). We note that the main goal of our learned Laplacian is not to simply improve classification and regression tasks. The main point is two fold: 1) to define an interpretable Laplacian that improves vanilla Laplacians on tasks, and , 2) to use the task for self-supervision for extracting the density, which is the main object of interest, as it can be used for knowledge extraction from graphs. We note that the Laplacian in Dasoulas et al is not interpretable. It is simply a linear combination of well known vanilla Laplacians.
> >
> > >**Q. Comparing density to degree**. Finally, for the learned 1/\rho, of the real-world networks, it would be interesting to see some plots and how they are different from let's say 1/\sqrt{deg}.
> >
> > **A**. We added a new figure to the revised paper to address this. Fig. 6 (Appendix A.2, p.14) compares the degree of nodes against the density learned to correct the GSO and the density learned to perform pooling for the AIDS dataset. From the plot we notice that the degree cannot predict the density. Indeed, the sampling density at nodes with the same degree can have different values.
> >
> > >**Q. Clarity, Quality, Novelty And Reproducibility**. This work in general could be more clearly written, starting from motivating properly what this work is trying to do and why.
> >
> > A. With respect to clarity, we refer to the answer to the last question.
> >
> > >**Q. Goal of the Laplacian normalization**. For example, do we just want to derive a proper normalization of the adjacency matrix in a way that improves classification performance, or we want to explain in a theoretical grounded way why this normalization is necessary?
> >
> > **A**. We want to normalize the Laplacian in an interpretable way that allows to derive knowledge from the graph in a theoretically grounded way.
> > Since “we see GSOs as randomly sampled from kernel operators defined on underlying geometric spaces” (Section 2.1, p.3)  and since “we are interested in GSOs that can be computed directly from the graph structure, without explicitly knowing the underlying continuous kernel and density” (Section 2.2, p. 4), we prove in Prop. 1, Section 2.2, p. 5  “ that if we are given graph that was sampled from a neighborhood model, and we know (or have an estimate of) the sampling density at every node of the graph, then we can compute a GSO according to (4) that is guaranteed to approximate a corresponding unknown metric-probability Laplacian. The next goal is hence to estimate the sampling density from a given graph.” Prop. 1 states the importance of the normalization to guarantee convergence. The experiments in Section 3 are then used to show that the proposed method for learning the underlying density with self-supervision can indeed learn an important feature since it enhances performance for the task at hand.
> >
> > >**Q. Structure and streamlining the paper**. Also, I think section 2 is quite verbose and each subsection does not seamlessly build on what is discussed in the previous one. E.g., section 2.2 starts by discussing again eq 1 with little reference to what was discussed right before (in deriving eq 2).

---

> > > ### Author Response · Authors · 2022-11-18
> > > **Response to Reviewer swjz (3/3)**
> > >
> > >
> > >
> > >
> > > **A**. We extensively revised and streamlined the paper to improve clarity (see last question). The changes in Section 2 are made to guarantee the development of the theory is as cohesive as possible.
> > > - In Section 2.1, p. 3 “ we see GSOs as randomly sampled from kernel operators defined on underlying geometric spaces.”
> > > - In section 2.2, p. 4 we focus on GSOs that can be computed directly from the graph structure:  “According to (2), a GSO $\mathbf{L}$ can be directly sampled from the metric-probability Laplacian $\mathcal{L}$. However, such an approach would violate our motivating guidelines, since we are interested in GSOs that can be computed directly from the graph structure, without explicitly knowing the underlying continuous kernel and density.”
> > > - In Section 2.3, p. 5 we give “a first rough estimate of the sampling density in a special case” via the degree of a node.
> > > - In Section 2.4, p. 5 we note that “When designing a method to estimate the sampling density from the graph, the degree is not a sufficient input parameter. The reason is that the degree of a node has two main contributions: the sampling density and the neighborhood radius.” Therefore, in the same section we focus on geometric graphs with hubs, for which the problem becomes manageable.
> > > - In Section 2.5, p. 6 “we propose a strategy to assess the sampling density $\pmb{\rho}$” in a self-supervised fashion. “Since in real-world scenarios the ground-truth density is not known, we train $\Theta$ [the density estimator, A/N] in a self-supervised manner. In this context, we choose a task on a real-world graph $\mathcal{G}$ and we solve it by means of a  graph neural network $\Psi$, referred to as \emph{task network}. Since we want  $\Psi$ to depend on the sampling density estimator $\Theta$, we define $\Psi$ as a spectral graph convolution network based on the non-uniform geometric GSO  $\mathbf{L}_{\mathcal{G},\Theta(\mathcal{G})}$”
> > >
> > > >**Q. Additional experiments**. In terms of quality more could be done in the experimental section with respect to comparing with other methods for learning parameterized graph shift operators, or a discussion with other latent models that try to learn graphs and assume different levels of popularity for each node.
> > >
> > > **A**. We added experiments on link prediction (Fig. 2, Section 3.1, p. 7 and Fig. 7, Appendix A.3, p. 15) and node classification (Fig. 3, Section 3.2, p. 8)  for three new datasets (Amazon Computers, Amazon Photo and FacebookPagePage).
> > > In link prediction, we now compare also against an auto-encoder with MLP decoder. We chose this new autoencoder to have approximately the same number of parameters as our geometric graph with hubs auto-encoder, for fair comparison (see Tab. 1, Appendix A.3, p. 15). Fig. 2 (Section 3.1, p. 7) and Fig. 7 (Appendix A.3, p. 15) show that our geometric graph with hubs auto-encoder is more accurate. Apart from accuracy, the main point is that our auto-encoder is also more interpretable, which allows us to use it for extracting knowledge from graphs (see Section 3.4 pp. 8-9).
> > >
> > >
> > > >**Q. Summary Of The Review**. I believe this work has some potential, though certain things need to be addressed. Mostly related to clarity, better motivating this work and focussing on the exact question/problem it tries to answer/solve. This will drive both the theoretical explanation and empirical evaluation in a more clear and concrete way.
> > >
> > > **A**. We thank the reviewer for the constructive criticism and suggestions. We believe that we have addressed the points raised by the reviewer, which made the paper significantly better.
> > > With respect to clarity,
> > > - We rewrote  “Our Contribution” (Section 1.1, pp.2-3) to spell out our contribution clearly and concisely.
> > > - We now explain in Section 2.1 p. 3 what a GSO is, why it is important in graph signal processing and how they are used to define spectral graph convolutional neural networks
> > > - The former sections “Non-Uniform Geometric Metric-Probability Laplacian” and “Non-Uniform Geometric Graph Laplacian” have been merged into Section 2.2, pp. 4-5 to give a more coherent development of the theory. We clarified how a nonuniform geometric GSO could be computed directly from the graph structure and the estimated sampling density.
> > > - The former Section “Learning the Sampling Density for Geometric Graphs with Hubs” has been split into “Geometric Graphs with Hubs” (Section 2.4, pp. 5-6) and “Learning the Sampling Density” (Section 2.5, p. 6) in order to clearly separate the density estimation method from the model of the graph.
> > > - We better explain how a non-uniform geometric GSO (Def. 4, Section 2.2, p. 4) can be computed without sampling directly the continuous Laplacian which is unknown
> > > - We now highlight the importance of Prop. 1, Section 2.2, p. 5
> > > - We now motivate our model of geometric graphs with hubs (Section 2.4, p. 6)
> > > We hope that the reviewer agrees that the paper is much improved, and will consider raising their score accordingly.

---

> > > > ### Comment · Reviewer_swjz · 2022-11-28
> > > > **Thanks for the response**
> > > >
> > > > I would like to thank the authors for their detailed response. In particular, I believe that the scope of the paper is now more clearly stated, while the contributions of it better exposed. Thus, I have decided to raise my score.

---

### Author Response · Authors · 2022-11-18
**Common Response**

We thank the reviewers for their thorough and insightful remarks. We fully implemented their comments in the revised version of the paper, which we believe improved the paper significantly.

**The main changes:**

 1) The former section “Related Work” and part of the former “Our Contribution“ is now in the “Introduction” (Section 1, pp. 1-2), which we shortened and more focused on the topic of our paper in order to streamline the paper.
 2) The “Our Contribution” (Section 1.1, pp. 2-3) has been rewritten to clearly state our contribution. We structured the current “Our Contribution” to highlight the main difference between our approach and traditional ones (paragraph 1), the difference between our NuG model and standard geometric graphs (paragraph 2), and the novelty of our proposed method for learning the underlying sampling density.
3) In order to make the background clearer, we now recall in Section 2.1 p. 3 the definition of GSOs, explaining why they are important in graph signal processing and how they are used to define spectral graph convolutional neural networks.
4) The former sections “Non-Uniform Geometric Metric-Probability Laplacian” and “Non-Uniform Geometric Graph Laplacian” have been merged into Section 2.2, pp. 4-5 to streamline the paper and give a more coherent development of the theory. We clarified that our model comprises only symmetric unweighted graphs, and we explained more clearly how a nonuniform geometric GSO could be computed directly from the graph structure and the estimated sampling density.
5) The former Section “Learning the Sampling Density for Geometric Graphs with Hubs” has been split into “Geometric Graphs with Hubs” (Section 2.4, pp. 5-6 ) and “Learning the Sampling Density” (Section 2.5, p. 6) in order to clearly separate the density estimation method from the model of the graph. We believe that the density estimation method is now written more clearly.
6) We added motivation for the geometric graph with hubs model, from a modeling point of view, showing that the assumptions are indeed natural when describing natural graphs (Section 2.4, p. 6).
7) We now explain the significance of Prop. 1 more clearly (Section 2.2, p.5).
8) We added the proof of Lemma 1 to Appendix D, p. 21.

**New experiments:**

1) We added new experiments that in the previous version were missing. Specifically, we used three new datasets (Amazon Computers, Amazon Photo, FacebookPagePage) for link prediction and node classification tasks Figs. 2-3, 7 show the performances on such datasets on link prediction, node classification respectively. They also show that our methods work also on datasets larger than the common citation networks (Cora, Citeseer, Pubmed). In order to have a fair comparison, we tested our geometric graph with hubs auto-encoder against an auto-encoder where the decoder is a multi-layer perceptron (MLP). We chose this new autoencoder to have approximately the same number of parameters as our geometric graph with hubs auto-encoder (see Tab. 1, Appendix A.3, p. 15).  Our geometric graph with hubs auto-encoder can represent these more complex networks better than the other methods and have accurate and consistent performances across datasets.
2) We added Fig. 6, Appendix A.2, p. 14 to show a comparison between the degree and the learned density via our proposed self-supervised method. From the plot we notice that the degree cannot predict the density. Indeed, the sampling density at nodes with the same degree can have different values.
3) We added Tab. 1 (Appendix A.3, p. 15) to compare the real-world graphs in terms of the number of parameters of their corresponding graph auto-encoders., to show that the geometric graph auto-encoder has a comparable number of learnable parameters to the auto-encoder with inner product decoder, and that our geometric graph with hubs auto-encoder has approximately the same number of parameters as the auto-encoder with MLP decoder.

---

### Decision · Program_Chairs · 2023-01-20

**Decision:**

Accept: poster

**Justification For Why Not Higher Score:**

The paper is quite technical with a good need for the appropriate technical background, and is not suitable for an oral or spotlight.

**Justification For Why Not Lower Score:**

A refreshing direction for ICLR, and good steps put forward toward the rigorous understanding of non-uniform geometric graphs with unknown sample density.

**Metareview: Summary, Strengths And Weaknesses:**

Geometric graphs have long been studied: a natural random model is that the nodes are sampled from some distribution over a metric space, along with some neighborhood radius alpha(u) for each node u---nodes u and v are then considered adjacent iff the distance d(u,v) between u and v in the metric space is at most max{alpha(u), alpha(v)). Much of the literature focuses on uniformly-random sampling and constant alpha. However, social networks with communities and hubs (such as superspreaders) easily necessitate non-uniform sampling and varying neighborhood radius. Under this model, a graph shift operator (GSO) can be viewed as a discretization of the continuous Laplacian. In order for this GSO to approximate the continuous Laplacian, the adjacency matrix needs to be normalized according to the unknown sampling density: the sampling density is estimated using a neural network and self-supervised learning.

The non-uniform model considered here seems reasonable for real-world graphs. Experiments on synthetic datasets suggest that this approach can indeed well-approximate the true sampling density. Additional experiments show improvement in classification tasks, while learning the density values can be used to assign importance scores to nodes. Fundamentally, several experiments here show how the density estimation improves the performance of classical graph-convolutional networks.

The authors are asked to incorporate the reviewer comments on writing.


**Note From Pc:**

if the above contains the word "oral" or "spotlight" please see: "oral" presentation means -> notable-top-5% and "spotlight" means -> notable-top-25%. As stated in our emails, we are disassociating presentation type from AC recommendations

**Summary Of Ac-Reviewer Meeting:**

N/A